# Oxydifficidin, a potent *Neisseria gonorrhoeae* antibiotic due to DedA-assisted uptake and ribosomal protein RplL sensitivity

**Jingbo Kan[1,2,3], Adrian Morales-Amador[1], Yozen Hernandez[1], Melinda A Ternei[1], Christophe Lemetre[1], Logan W MacIntyre[1], Nicolas Biais[2,3,4]\*, Sean F Brady[1]\***

[1]Laboratory of Genetically Encoded Small Molecules, The Rockefeller University, New York, United States; [2]Graduate Center, City University of New York, New York City, United States; [3]Brooklyn College, City University of New York, New York, United States; [4]Laboratoire Jean Perrin, UMR 8237 Sorbonne Université/CNRS, Paris, France

**\*For correspondence:**
nicolas@mechano-micro-biology.org (NB);
sbrady@rockefeller.edu (SFB)

**Competing interest:** The authors declare that no competing interests exist.

## eLife Assessment

Kan et al. report the discovery of a Bacillus amyloliquifaciens strain that kills Nerisseria gonorrhoeae via oxydifficidin which targets ribosomal proteins. Resistance occurred via mutation in the DedA flippase to influence oxydifficidin uptake. The overall mechanism of action is well described making this an **important** study with implications for combating clinical antibiotic resistance. The evidence presented is **convincing** due to rigour employed in the methodological approach. The authors should consider performing a more comprehensive genetic analyses of DedA and RplL in this clinically relevant strain. This work will be of broad interest to microbiologists and synthetic biologists.

**Abstract** Gonorrhea, which is caused by *Neisseria gonorrhoeae*, is the second most reported sexually transmitted infection worldwide. The increasing appearance of isolates that are resistant to approved therapeutics raises the concern that gonorrhea may become untreatable. Here, we serendipitously identified oxydifficidin as a potent *N. gonorrhoeae* antibiotic through the observation of a *Bacillus amyloliquefaciens* contaminant in a lawn of *N. gonorrhoeae*. Oxydifficidin is active against both wild-type and multidrug-resistant *N. gonorrhoeae*. Its potent activity results from a combination of DedA-assisted uptake into the cytoplasm and the presence of an oxydifficidin-sensitive ribosomal protein L7/L12 (RplL). Our data indicate that oxydifficidin binds to the ribosome at a site that is distinct from other antibiotics and that L7/L12 is uniquely associated with its mode of action. This study opens a potential new avenue for addressing antibiotic resistant gonorrhea and underscores the possibility of identifying overlooked natural products from cultured bacteria, particularly those with activity against previously understudied pathogens.

## Introduction

Gonorrhea is the second most reported sexually transmitted infection worldwide, its causative agent is the bacterium *Neisseria gonorrhoeae*. According to the World Health Organization (WHO) approximately 82.4 million new adult gonorrhea infections occurred globally in 2020 (*WHO, 2019*). The high dose (500 mg) of the cephalosporin ceftriaxone is currently the only recommended therapy for treating gonorrhea infections in the USA (*St Cyr et al., 2020*). The growing instances of drug resistant 'superbugs', together with the limited clinical treatment options, underscore the urgent need

for additional antibiotics that target *N. gonorrhoeae* (*Unemo et al., 2012*; *Ohnishi et al., 2011*; *Day et al., 2022*; *Derbie et al., 2020*; *Sawatzky et al., 2022*; *Shaskolskiy et al., 2022*). The characterization of antibacterial active natural products inspired the development of both ceftriaxone and azithromycin (*Richards et al., 1984*; *Bakheit et al., 2014*). Bacterial natural products have been a key source of antibiotics with diverse modes of action and the most fruitful source of therapeutically useful antibiotics (*Newman and Cragg, 2020*; *Dias et al., 2012*; *Lewis, 2020*). Here, we describe the serendipitous identification of the natural product oxydifficidin (*Wilson et al., 1987*; *Zimmerman et al., 1987*) as a potent *N. gonorrhoeae* active antibiotic and show that this activity arises from a combination of DedA flippase-assisted uptake and ribosomal protein L7/L12 (RplL) sensitivity. Oxydifficidin provides a new therapeutic lead structure for addressing the growing problem of antibiotic resistant gonorrhea. Over the last century, bacteria were extensively examined for antibiotic production in screens that often focused on a small number of pathogens. This study suggests that reexamining cultured bacteria for antibiotics active against today's emerging pathogens may be fruitful as metabolites with specific potent activity against historically less problematic pathogens may have been overlooked.

## Results

### Oxydifficidin isomers selectively and potently inhibit *N. gonorrhoeae*

In our day to day experiments, we regularly use agar plates containing lawns of pathogenic bacteria. During these experiments, we often find random environmental bacteria growing on these plates. On one lawn of *N. gonorrhoeae* we observed an environmental contaminant that was surrounded by a zone of growth inhibition suggesting that it produced an anti-*N. gonorrhoeae* metabolite (*Figure 1a*). When we screened this contaminant for antibacterial activity against lawns of other Gram-negative bacteria it did not produce a zone of growth of inhibition against any of the bacteria we tested (e.g., *Escherichia coli*, *Vibrio cholerae*, *Caulobacter crescentus*). Since antibiotics that preferentially inhibit the growth *N. gonorrhoeae* are rare, we looked at this contaminant in more detail. Sequencing of the contaminant's genome and genome clustering analysis (*Appendix 1—figure 1a*) revealed that it was most closely related to *Bacillus amyloliquefaciens*, which is a root-colonizing bacterium that is used as a biocontrol agent (*Ngalimat et al., 2021*). We named the anti-*N. gonorrhoeae* contaminant *B. amyloliquefaciens* BK.

To identify the biosynthetic gene cluster (BGC) responsible for the observed antibiosis we screened *B. amyloliquefaciens* BK transposon mutants for strains that no longer produced anti-*N. gonorrhoeae* activity. The sequencing of the non-producer strain revealed that it surprisingly contained four transposon insertions and one frame shift mutation (*Appendix 1—figure 1b*). The frame shift mutation and one transposon insertion were predicted to each disrupt unique BGCs. The transposon inserted into the bacillomycin (mycosubtilin) BGC, while the frame shift mutation was predicted to disrupt the (oxy)difficidin BGC. To determine which of these two BGCs was responsible for the *N. gonorrhoeae* activity we tested *B. amyloliquefaciens* strains with targeted disruptions of each BGC for activity against *N. gonorrhoeae* (*Koumoutsi et al., 2004*). Only disruption of the (oxy)difficidin BGC eliminated the anti-*N. gonorrhoeae* activity (*Appendix 1—figure 1c*). To confirm the identity of the *N. gonorrhoeae* active antibiotic we carried out a bioassay guided fractionation of *B. amyloliquefaciens* BK culture broth (*Wilson et al., 1987*; *Zimmerman et al., 1987*). High-resolution mass spectrometry (HRMS) and nuclear magnetic resonance (NMR) data from the major active peak we isolated were consistent with its being an oxydifficidin isomer (*Figure 1b*, *Appendix 1—figures 5–12*, and *Supplementary file 2*). Oxydifficidin contains a 27-carbon polyketide backbone that is cyclized through a terminal carboxylic acid and an oxidation at position 21. This hydrophobic core is phosphorylated at C16. Oxydifficidin occurs naturally as a collection of interconverting thermal isomers (*Figure 1b*). As reported previously, we observed an interconversion of isomers with the compound we purified from *B. amyloliquefaciens* BK cultures (*Wilson et al., 1987*; *Zimmerman et al., 1987*). All assays were performed using the mixture of interconverting oxydifficidin isomers we obtained from *B. amyloliquefaciens* BK cultures.

We tested oxydifficidin for activity against diverse bacterial pathogens. Oxydifficidin showed only weak activity against most pathogens, however we observed potent activity against *N. gonorrhoeae* (*Figure 1c*). When we examined other *Neisseria* species, we found that oxydifficidin was consistently more active against *Neisseria* than any of the other bacteria we tested. Among *Neisseria* spp., oxydifficidin was most active against *N. gonorrhoeae* underscoring a unique narrow spectrum of potent

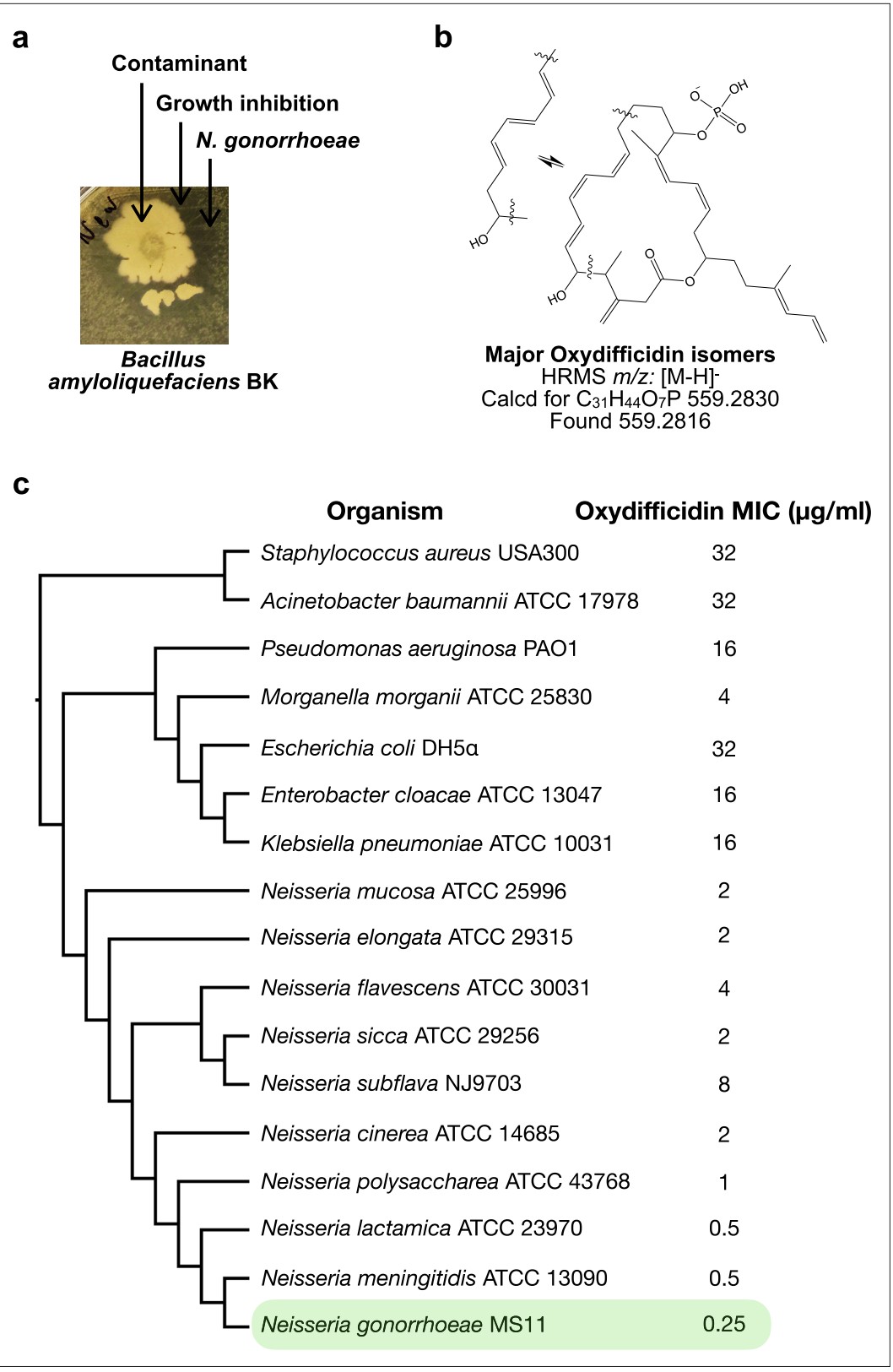

**Figure 1.** Oxydifficidin isomers inhibit the growth of *N. gonorrhoeae*. (**a**) Discovery of a contaminant (*Bacillus amyloliquefaciens* BK) that inhibited the growth of *N. gonorrhoeae*. (**b**) Example of known oxydifficidin isomers. (**c**) Minimum inhibitory concentration (MIC) of oxydifficidin against bacteria (*n* = 2). Genome-based phylogenetic tree was built by Genome Clustering of MicroScope using neighbor-joining method.

**Table 1.** Susceptibilities of *N. gonorrhoeae* to antibiotics.

| Clinically relevant antibiotic | MIC (µg/ml) | | | |
|---|---|---|---|---|
| | MS11 | H041 | AR#1280 | AR#1281 |
| Ceftriaxone | 0.125 | 1 | 1 | 1 |
| Azithromycin | 0.25 | 0.5 | 0.5 | 1 |
| Ciprofloxacin | 0.031 | 32 | 16 | 16 |
| Gentamicin | 8 | 8 | 8 | 8 |
| Tetracycline | 1 | 1 | 1 | 1 |
| Mode of action relevant antibiotic | | | | |
| Oxydifficidin | 0.25 | 0.125 | 0.125 | 0.125 |
| Ceftazidime | 0.125 | 16 | 1 | 8 |
| Ampicillin | 1 | 8 | 2 | >64 |
| Chloramphenicol | 4 | 4 | 4 | 4 |
| Rifampicin | 0.25 | 0.25 | 0.125 | 0.125 |
| Nalidixic acid | 16 | 16 | 16 | 16 |
| Irgasan | 0.5 | 0.5 | 0.5 | 0.5 |
| Vancomycin | >64 | >64 | 64 | >64 |
| Polymyxin B | >64 | >64 | 64 | >64 |
| Melittin | 2 | 2 | 1 | 2 |
| Nisin | 4 | 4 | 4 | 4 |
| Bacitracin | 4 | 4 | 1 | 4 |
| Daptomycin | >64 | >64 | >64 | >64 |

*n* = 2.

activity. A key issue with the current treatment of *N. gonorrhoeae* infections is the development of resistance to existing therapeutics. Resistance to the standard of care cephalosporins is particularly problematic. Oxydifficidin was more potent against *N. gonorrhoeae* MS11 than most other antibiotics we tested. Notably, unlike clinically used antibiotics such as ceftriaxone, azithromycin, and ciprofloxacin, oxydifficidin retained activity against all multidrug-resistant clinical isolates we examined (*Table 1*).

Oxydifficidin's structure is interesting as phosphorylated antibiotics, and moreover natural products in general, remain rare (*Petkowski et al., 2019*; *Cao et al., 2019*). Its structure together with its specific and potent activity against drug resistant *N. gonorrhoeae* suggested it might have a unique mode of action (MOA). This is appealing from the perspective of developing therapeutics that are capable of circumventing clinically problematic resistance mechanisms and therefore we focused on characterizing the mechanism of oxydifficidin's potent anti-*N. gonorrhoeae* activity.

## DedA assists the uptake of oxydifficidin into *N. gonorrhoeae*

As a first step to understanding the MOA of oxydifficidin, we raised resistant mutants by directly plating *N. gonorrhoeae* cultures on antibiotic containing plates (1 µg/ml, 4× minimum inhibitory concentration [MIC]) (*Figure 2a*). Out of the >1.5 × 10^10 cells we screened, only one resistant mutant appeared. Oxydifficidin's MIC against this mutant increased by eightfold (2 µg/ml). No increase in MIC was observed for any other antibiotic we tested (*Figure 2a*). Sequencing and comparison of this mutant's genome to the sensitive parent genome revealed a single mutation that introduced a frame shift in one of the three predicted *dedA* genes found in the *N. gonorrhoeae* MS11 (NCBI:txid528354 *dedA* NGFG_RS04905; *Jen et al., 2021*). To confirm that DedA was necessary for oxydifficidin's potent

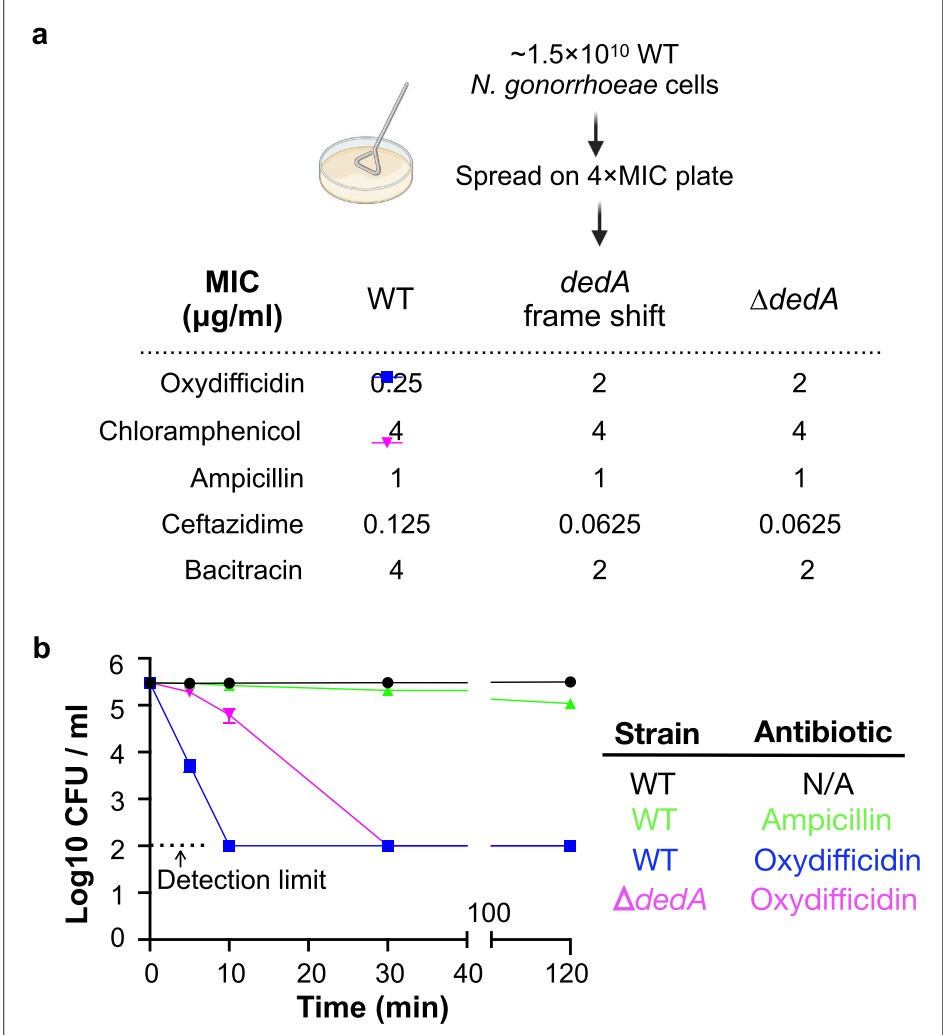

**Figure 2.** Oxydifficidin-resistant *N. gonorrhoeae* mutant development and corresponding susceptibilities. (**a**) Schematic representation of *N. gonorrhoeae* mutant development that identified *dedA*. Activity of different antibiotics against MS11 and *dedA* gene disrupted *N. gonorrhoeae* MS11. (**b**) Time-dependent antibiotic killing assay of *N. gonorrhoeae* strains. Each antibiotic was tested at 8× its minimum inhibitory concentration (MIC) for the specific strain being examined (MS11 Ampicillin: 8 µg/ml; MS11 Oxydifficidin: 2 µg/ml; MS11 Δ*dedA* Oxydifficidin: 16 µg/ml). Δ*dedA* indicates *N. gonorrhoeae* MS11 *dedA* deletion mutant (*n* = 3). Bars represent mean CFU values ± SD.

activity, we created a *dedA* deletion mutant in a clean *N. gonorrhoeae* background. This mutant showed the same eightfold increase in oxydifficidin's MIC, confirming that the deletion of *dedA* is sufficient to reduce oxydifficidin's potent activity. We also generated deletion mutants for two other predicted *dedA*-like genes, and the MIC of oxydifficidin for these mutants remained the same as for the *N. gonorrhoeae* MS11 wild-type (WT) strain. Interestingly, not only was *dedA* deficient *N. gonorrhoeae* less susceptible to oxydifficidin, oxydifficidin also kills this mutant more slowly (***Figure 2b***) than WT *N. gonorrhoeae* MS11. The *dedA* deletion mutant also showed an altered cell morphology with reduced membrane integrity and lower formation of microcolonies (***Appendix 1—figures 4***). A survey of 220 *N. gonorrhoeae* strains with high-quality assemblies in NCBI found no mutations in the DedA protein.

The DedA protein superfamily is highly conserved, with examples in almost every sequenced genome across all domains of life (***Doerrler et al., 2013***). DedA family members are predicted to be transmembrane proteins with still largely, poorly defined, functions. However, a few recent studies indicate that DedA homologs are flippases. They have been reported to flip phospholipids

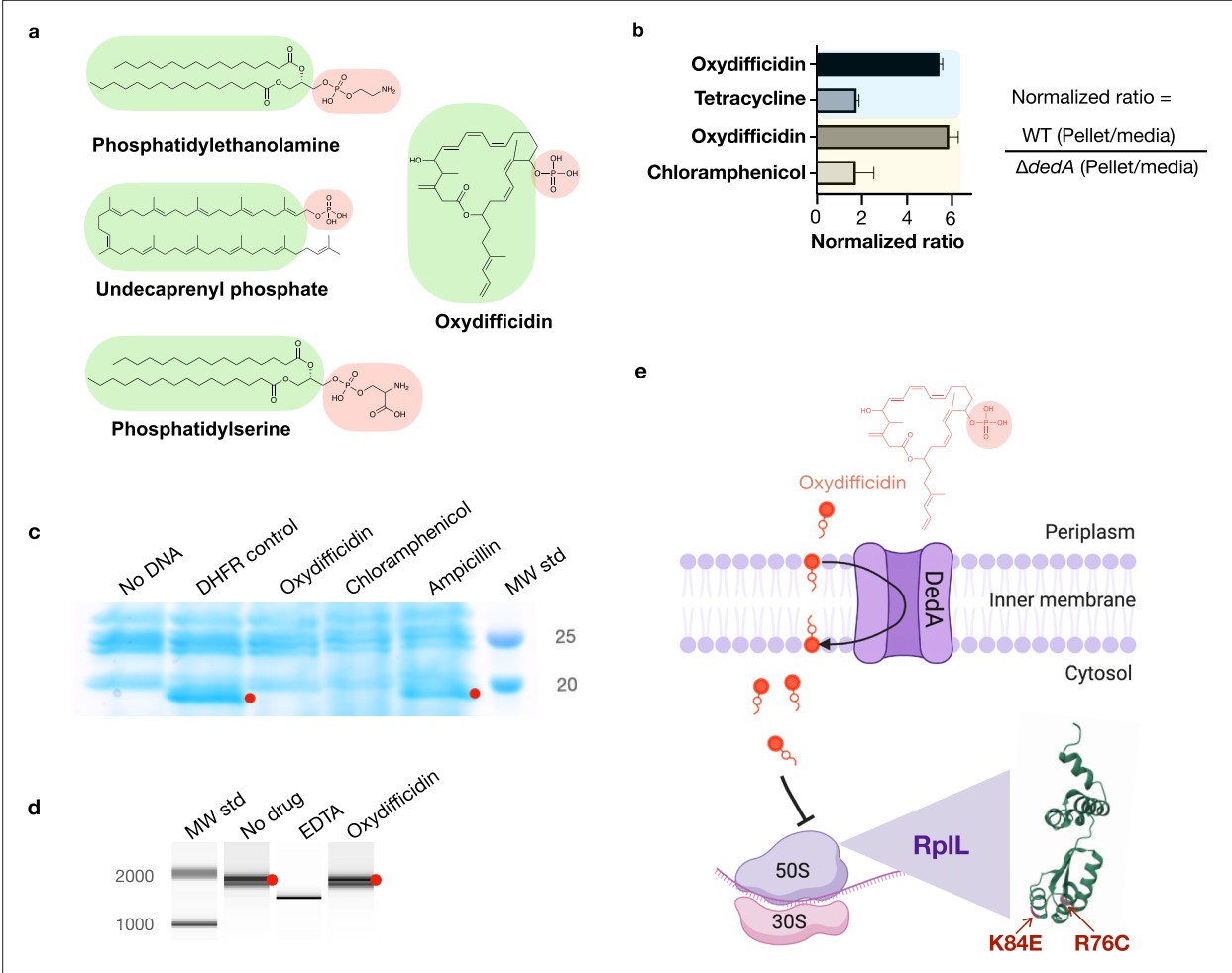

**Figure 3.** Oxydifficidin's anti-*N. gonorrhoeae* activity arises from a combination of DedA flippase-assisted uptake and ribosomal protein L7/L12 (RplL) sensitivity. (**a**) Structure of oxydifficidin compared to that of the known substrates for DedA homologs. (**b**) Comparison of antibiotic accumulation in MS11 and MS11 *dedA* knockout cells. Blue and yellow highlighted sections represent independent experiments (oxydifficidin and tetracycline: *n* = 2; oxydifficidin and chloramphenicol: *n* = 3). (**c**) In vitro coupled transcription/translation assay. The effect of oxydifficidin and other antibiotics on in vitro protein production using a coupled transcription/translation system was monitored by SDS–PAGE. Red dots indicate in vitro production of dihydrofolate reductase (18 kDa) from the *DHFR* gene. MW std: kDa molecular weight standard. (**d**) In vitro transcription assay. Red dots indicate in vitro production of a 1704-bp RNA from the *FLuc* gene. A reaction containing 20 mM of EDTA was used as an inhibition control. MW std: bp molecular weight standard. (**e**) Model explaining oxydifficidin's potent activity in *N. gonorrhoeae*. In this model DedA flips oxydifficidin across the inner membrane to assist its uptake and oxydifficidin then inhibits protein synthesis through either a direct or indirect interaction with L7/L12 (RplL). Two spontaneous mutations (K84E and R76C) in the RplL (L7/L12) protein were found to confer resistance to oxydifficidin. Created using BioRender.com. Bars represent mean ratio values ± SD.

The online version of this article includes the following source data for figure 3:

**Source data 1.** Original SDS–PAGE picture for *Figure 3c*.

**Source data 2.** Original SDS–PAGE picture for *Figure 3c*, labeled.

**Source data 3.** Original BioAnalyzer report for *Figure 3d*.

**Source data 4.** Original BioAnalyzer report for *Figure 3d*, labeled.

(phosphatidylethanolamine, phosphatidylserine) or phospholipid-like structures (C55-isoprenyl pyrophosphate) across prokaryotic and eukaryotic lipid bilayers (*Sit et al., 2023*; *Roney and Rudner, 2023*; *Huang et al., 2021*; *Li et al., 2021*). Although oxydifficidin is not a phospholipid, its overall structure resembles that of reported DedA protein substrates, especially C55-isoprenyl pyrophosphate (*Figure 3a*). Interestingly, among the two characterized bacterial family members, the DedA protein associated with oxydifficidin potency is most closely related to the C55-isoprenyl pyrophosphate flippase YghB from *V. cholerae* (*Appendix 1—figure 2*; *Sit et al., 2023*). As oxydifficidin's activity was

not completely abrogated in the *dedA* knockout we postulated that DedA was not the direct target of oxydifficidin, but it instead acted to increase oxydifficidin's potency in *N. gonorrhoeae*. The structural similarity between oxydifficidin and the known substrates of DedA homologs led us to explore the possibility that DedA was responsible for assisting with oxydifficidin uptake into *N. gonorrhoeae*.

We examined the effect of DedA on antibiotic accumulation by comparing the amount of compound found in the cell pellet collected from antibiotic treated cultures of WT and *dedA* knockout *N. gonorrhoeae* strains (**Figure 3b**). In the case of oxydifficidin we saw six times more antibiotic in cells from WT cultures than from *dedA* deletion strain cultures. For the other antibiotics we tested (tetracycline and chloramphenicol) this ratio was less than two. Based on DedA homologs flipping phospholipid-like structures across a lipid bilayer our data is consistent with DedA flipping oxydifficidin across the inner membrane to increase its cytoplasmic concentration and in turn increase its potency against *N. gonorrhoeae*. A DedA-assisted uptake mechanism could also explain the slower rate of killing we observed for oxydifficidin against *dedA* deficient *N. gonorrhoeae* compared to WT *N. gonorrhoeae*. While we cannot definitely rule out the possibility that DedA accumulation of oxydifficidin in the membrane could also have a direct toxicity effect, we did not detect any cell lysis or membrane depolarization at even 100 times oxydifficidin's MIC (**Appendix 1—figure 3**).

## Oxydifficidin inhibits protein synthesis by interacting with RplL

To look for an intracellular target of oxydifficidin we carried out a second round of resistant mutant screening. In this case, we used the *N. gonorrhoeae dedA* deletion strain and searched for colonies with an even higher tolerance to oxydifficidin. From ~1 × 10$^{10}$ cells plated on 8 µg/ml oxydifficidin (4× MIC for *N. gonorrhoeae dedA* deletion strain) we identified 12 resistant mutants. In each case, the oxydifficidin MIC increased to 16 µg/ml. Sequencing of these strains revealed that each contained a point mutation in the gene encoding for the large ribosomal protein(s) L7/L12 (*rpl*L). Eleven strains contained the same R76C mutation and one contained a K84E mutation (**Supplementary file 1**). These two mutations were not found in the survey of the same collection of *N. gonorrhoeae* strains used to look for DedA mutations.

To determine if mutations in the *rpl*L gene alone were sufficient to confer oxydifficidin resistance, we created an RplL R76C mutant in a WT *N. gonorrhoeae* background (i.e., non *dedA* deletion). This mutant exhibited an eightfold increase in oxydifficidin's MIC (2 µg/ml) compared to the parent strain, confirming that the R76C mutation in L7/L12 alone was sufficient to increase the MIC of oxydifficidin. Ribosomal proteins L7 and L12 have the same sequence, however L12 has an N-terminal acetylation (**Diaconu et al., 2005**). L7/L12 is part of the L10/L7 stalk of the large (50S) subunit of the bacterial

**Table 2.** Activity of antibiotics against *N. gonorrhoeae rpl*L mutant.

| Antibiotic | MIC (µg/ml) | |
| --- | --- | --- |
| | MS11 | MS11 RplL_R76C (Engineered) |
| Oxydifficidin | 0.25 | 2 |
| Ampicillin | 1 | 1 |
| Ceftazidime | 0.125 | 0.125 |
| Bacitracin | 4 | 4 |
| Ribosome-targeting antibiotic | | |
| Chloramphenicol | 4 | 4 |
| Spectinomycin | 16 | 16 |
| Tetracycline | 1 | 1 |
| Erythromycin | 0.5 | 0.5 |
| Gentamicin | 8 | 8 |
| Avilamycin | 4 | 4 |
| Thiostrepton | 0.125 | 0.125 |

*n* = 2.

ribosome and is critical to a number of processes including GTP hydrolysis (*Diaconu et al., 2005*; *Carlson et al., 2017*).

The appearance of resistance mutations in *rplL* suggested that oxydifficidin inhibited protein synthesis, which would be consistent with isotope feeding studies performed with difficidin and *E. coli* (*Zweerink and Edison, 1987*). We initially tested this hypothesis in vitro using a coupled transcription/translation system. While this reaction mixture contained all the components necessary to produce a protein from DNA, no protein was produced in the presence of oxydifficidin (*Figure 3c*). Using a coupled system, it was not possible to distinguish between inhibition of RNA or protein synthesis, and therefore, to rule out inhibition of transcription, we next looked directly at RNA synthesis in vitro. In this case we saw no inhibition of RNA synthesis, even at the highest oxydifficidin concentration we tested (13.4 µg/ml) (*Figure 3d*). Taken together these two experiments indicate that oxydifficidin inhibits translation but not transcription and are consistent with *rplL* mutations providing resistance to oxydifficidin.

To the best of our knowledge *rplL* mutations have not been previously associated with antibiotic resistance and no characterized antibiotics have been found to bind L7/L12. When we screened for cross resistance, the L7/L12 R76C mutation did not confer resistance to any other antibiotics we tested (*Table 2*). These included ribosome targeting antibiotics with diverse binding sites. Antibiotics that bind different regions of the 30S decoding center, including tetracycline, gentamicin, and spectinomycin showed no increase in MIC with the L7/L12 R76C mutation (*Lin et al., 2018*; *Paternoga et al., 2023*). Similarly, the L7/L12 R76C mutation did not confer resistance to chloramphenicol or erythromycin, which interact with distinct regions of the 50S peptidyl transferase center (*Lin et al., 2018*; *Paternoga et al., 2023*). Antibiotics that bind away from these two hot spots also showed no cross resistance (*Lin et al., 2018*; *Paternoga et al., 2023*; *Polikanov et al., 2018*). The activity of avilamycin, which binds in a unique site in the 50S subunit at the entrance to the A-site tRNA accommodating corridor was also not affected by the L7/L12 R76C mutation (*Arenz et al., 2016*). In the case of thiostrepton A (*Harms et al., 2008*), which targets the L11 protein GTPase-associated center and is part of the only class of antibiotics known to bind the ribosome in the vicinity of L10/L7 stalk, we also observed no change in MIC for *N. gonorrhoeae* with the L7/L12 R76C mutation. The inability of the L7/L12 R76C mutation to confer cross resistance to known antibiotics suggests that oxydifficidin binds to a different site on the ribosome and that L7/L12 is uniquely associated with oxydifficidin's MOA.

To determine whether the DedA and RplL mutations we observed were specific to oxydifficidin resistance in *N. gonorrhoeae*, we selected for resistant mutants using two other *Neisseria* species: *Neisseria subflava* and *Neisseria cinerea*. Both strains were natively less susceptible to oxydifficidin than *N. gonorrhoeae*. In the case of *N. cinerea* all resistance mutants contained the RplL K84E mutation that we first observed in *N. gonorrhoeae,* and in the case of *N. subflava* all resistance mutants contained mutations in *dedA* (*Supplementary file 1*). Other *Neisseria* species likely show higher MICs than *N. gonorrhoeae* because of either the absence of DedA-assisted uptake or reduced RplL protein oxydifficidin sensitivity. The model that arises from our studies is that high oxydifficidin sensitivity results from a combination of DedA-assisted oxydifficidin uptake into the cytoplasm and the presence of an oxydifficidin-sensitive RplL protein (*Figure 3e*). *N. gonorrhoeae* is uniquely sensitive to oxydifficidin because it contains both. DedA is critical to oxydifficidin's potent activity as it not only sensitizes *N. gonorrhoeae* to oxydifficidin but also decreases its kill time, both of which are appealing from a clinical development perspective.

## Discussion

Almost 60% of clinically approved antibiotics target the ribosome (*Zhang et al., 2021*). Although it is the most common target for natural product antibiotics, most of these molecules inhibit the ribosome by binding to only a small number of sites (*Lin et al., 2018*; *Poehlsgaard and Douthwaite, 2005*). Clinically approved antibiotics generally inhibit protein synthesis by binding in either the ribosomal decoding center, peptidyl transfer center, or the peptide exit tunnel. Targeting unexploited sites of the ribosome is considered a key step to developing next generation antibiotics that can circumvent the existing antibiotic resistance mechanisms. Our data suggest that oxydifficidin has a distinct binding site in the ribosome compared to other clinically used antibiotics making it an appealing therapeutic candidate. In our resistance mutant screening experiments with *N. gonorrhoeae* the frequency of mutations in either *dedA* or *rplL* was quite low ($\sim 10^{-9}$). The *dedA* deletion

mutant exhibited altered cell morphology, characterized by diminished membrane integrity and reduced micro-colony formation (*Appendix 1—figure 4*), indicating that it should show reduced pathogenesis and fitness, and, as a result, not accumulate in a clinical setting, which adds to the therapeutic appeal of oxydifficidin.

While most natural product research efforts are focused on identifying novel chemical entities, the work presented here suggests that reexamining old sources through the lens of today's most important pathogens may also be a productive approach for identifying therapeutically appealing antibiotics. Increasing rates of antibiotic resistance present a significant clinical threat as they have the potential of rendering gonorrhea infections untreatable. The recent appearance of *N. gonorrhoeae* 'superbugs' with high-level resistance to all currently recommended drugs treating gonorrhea infections, is particularly concerning (*Ohnishi et al., 2011*; *Day et al., 2022*). One appealing feature in the original reports of the discovery of oxydifficidin was its activity in an animal model, a key hurdle for many natural products (*Zimmerman et al., 1987*). Unfortunately oxydifficidin's clinical significance was limited because of the weak activity it exhibited against most bacterial pathogens. Our identification of oxydifficidin as being specifically potent against *N. gonorrhoeae* including multidrug-resistant *N. gonorrhoeae*, provides a potential new path forward for this structurally interesting natural antibiotic.

## Materials and methods

### Bacterial strains and cultivation

Neisseria strains were grown at 37°C with 5% $CO_2$ in GCB medium (*Freitag et al., 1995*) (VWR CA90002-016). *B. amyloliquefaciens* BK was isolated from a lab agar plate, *B. amyloliquefaciens* FZB42 WT and various mutant strains were provided by Bacillus Genetic Stock Center. All Bacillus strains were grown at 30°C in Luria-Bertani (LB) medium. *E. coli* DH5α, *Klebsiella pneumoniae* ATCC 10031, *Enterobacter cloacae* ATCC 13047, *Acinetobacter baumannii* ATCC 17978, *Pseudomonas aeruginosa* PAO1, *Morganella morganii* ATCC 25830, and *Staphylococcus aureus* USA300 were grown at 37°C in Tryptic Soy Broth (TSB).

### Crude extraction and disc diffusion test

Bacillus strains were grown overnight at 30°C on LB agar. 3 ml of methanol was added to the top of the plate and spread uniformly by gentle shaking. The supernatant was then collected, centrifuged at 15,000 rpm for 2 min and filtered through a 0.2-μm cellulose acetate membrane (VWR). Then, 10 μl of the crude extract was added onto a paper disc (VWR) and after 5 min drying the paper disc was transferred on top of GCB agar plate lawned (swabbed from a single colony) with *N. gonorrhoeae* cells. The plate was incubated at 37°C with 5% $CO_2$ overnight and the inhibition zone was visually inspected.

### Fermentation of *B. amyloliquefaciens* BK

A single colony of *B. amyloliquefaciens* BK was grown in 250 ml LB in a 1-l flask at 30°C on a rotary shaker (200 rpm) overnight, then 50 ml of the overnight culture was transferred to 1 l modified Landy medium (20 g glucose, 5 g glutamic acid, 1 g yeast extract, 1 g $K_2HPO_4$, 0.5 g $MgSO_4$, 0.5 g KCl, 1.6 mg $CuSO_4$, 1.2 mg $Fe_2(SO_4)_3$, 0.4 mg $MnSO_4$ per 1 l deionized $H_2O$, pH adjusted to 6.5 before autoclaving) in a 2.8-l triple baffled flask and fermented at 30°C on a rotary shaker (200 rpm) for 3 days.

### Genomic sequencing and annotation of *B. amyloliquefaciens*

Genomic DNA from *B. amyloliquefaciens* BK WT and transposon mutant Tn5-3 was isolated using PureLink Microbiome DNA purification kit (Invitrogen) according to the manufacturer's instructions. The *B. amyloliquefaciens* BK WT genome was assembled by mapping its sequencing data onto the annotated genome of *B. amyloliquefaciens* FZB42 using Geneious Prime. Differences in the mutant strain Tn5-3 were identified by mapping its sequencing data onto the assembled *B. amyloliquefaciens* BK WT genome. The mutated genes were then annotated using NCBI BLAST. The oxydifficidin BGC was annotated using the antiSMASH online server.

## Isolation of oxydifficidin

The isolation process followed the published protocol (*Wilson et al., 1987*) with adjustments. The fermentation culture (5 l) was centrifuged at 4000 rpm for 10 min and the supernatant was acidified to pH 3.0 before extraction with 5 l of ethyl acetate. The crude extract was concentrated under reduced pressure (60 mBar) on a rotary evaporator to a final volume of about 20 ml. The concentrated sample was then loaded onto Diaion HP-20 resin (Sigma-Aldrich) and the resin was then washed by 500 ml of water, 500 ml of methanol–water (3:7) and 500 ml of methanol–water (7:3). The oxydifficidin rich cut was then obtained by eluting the resin with 500 ml of methanol. This rich cut was concentrated and absorbed onto DEAE-Sephadex A-25 (Cl⁻) resin (Sigma-Aldrich). The resin was washed with 250 ml of methanol–water (2:8), 500 ml of methanol–water (9:1), and a rich cut was then obtained by eluting the resin with 100 ml methanol–water (2:8) with 3% ammonium chloride. The rich cut from A-25 resin was then diluted with 300 ml of water and adsorbed onto Amberlite XAD-16N resin (Sigma-Aldrich) and washed successively with 500 ml of water, 250 ml of 0.075 M $K_2HPO_4$ (pH 7), and 250 ml of methanol–water (2:8), followed by elution with 100 ml of methanol. This oxydifficidin-containing eluate was diluted with 100 ml of 0.075 M $K_2HPO_4$ (pH 7) and applied to a RediSep Rf C18 column (Teledyne ISCO, 100 Å, 20–40 μm, 300 ± 50 m²/g) with a reverse-phase linear gradient system (5–100% 0.075 M $K_2HPO_4$/MeOH, 45 min). Fractions containing oxydifficidin was monitored by anti-*N. gonorrhoeae* activity and further purified by semipreparative HPLC using a reverse-phase C18 column (Waters,130 Å, 5 μm, 10 mm × 150 mm) with an isocratic system (0.075 M $K_2HPO_4$/MeOH (32:68)). The purified oxydifficidin was then desalted by Amberlite XAD-16N resin (Sigma-Aldrich).

## Electrospray Ionization LC–MS and NMR analysis

LC–HRMS data were acquired using a SCIEX ExionLC HPLC system coupled to an X500R QTOF mass spectrometer (The Rockefeller University). The system was equipped with a Phenomenex Kinetex PS C18 100 Å column (150 mm × 2.1 mm, 2.6 μm) and operated with SCIEX OS v.2.1 software. The following chromatographic conditions were used for LC–HRMS: 5% B to 2.0 min; 5–95% B from 2.0 to 30.0 min; 95% B from 30.0 to 37.0 min; 95–5% B from 37.0 to 38.0 min; and 5% B from 38.0 to 45.0 min (flow rate of 0.25 ml/min, 3 μl injection volume, A/B: water/acetonitrile, supplemented with 0.1% (vol/vol) formic acid. Both electrospray ionization (ESI) modes Full HRMS spectra were acquired in the range $m/z$ 100–1000, applying a declustering potential of 80 V, collision energy of 5 V, source temperature of 500°C and a spray voltage of 5500 V. A maximum of seven candidate ions from every Full HRMS event were subjected to Q2-MS/MS experiments in the range $m/z$ 60–1000, applying a collision energy of 35 ± 10 V for both ESI modes. Full HRMS and the most intense MS/MS spectra were analyzed with MestReNova software (14.3.0). NMR spectra were acquired on a Bruker Avance DMX 800MHz spectrometer equipped with cryogenic probes (New York Structural Biology Center). All spectra were recorded at 298 K in $CD_3OD$–$D_2O$ (1:1). Chemical shift values are reported in parts per million (ppm) and referenced to residual solvent signals: 3.31 ppm (1H) and 49.0 ppm (C). Spectra analysis and visualization were carried out in TopSpin (3.6.0) and MestReNova (14.3.0).

## MIC assay

Neisseria strains were grown on GCB agar plate overnight, cells were collected by a polyester swab tip (Puritan) and resuspended in 1 ml GCB medium followed by a 1 in 5000-fold dilution. The stock solution of antibiotics was serial diluted twofold in a 96-well plate (Thermo Fisher Scientific) containing GCB liquid medium, then an equal amount of diluted bacterial culture was added to each well and mixed by pipetting. The plates were incubated at 37°C with 5% $CO_2$ for 16 hr. *E. coli*, *K. pneumoniae*, and *M. morganii* strains were grown at 37°C in TSB medium overnight. The stock solution of antibiotics was serial diluted twofold in a 96-well plate containing TSB agar medium, then $10^4$ cells were spotted on each well. The plates were incubated at 37°C for 16 hr. The top and bottom rows of plates contained an equal volume of media alone to prevent edge effect. The last column did not contain antibiotic to serve as a control for cell viability. The MIC of antibiotics was determined by visual inspection as the concentration in the well that prevents bacterial growth compared to control wells. The MIC assays were performed in duplicate.

## Phylogenetic tree building

The phylogenetic trees for bacterial genomes were built using Genome Clustering function with default parameters in MicroScope (*Vallenet et al., 2020*). The bacterial DedA family protein sequences were downloaded from a previous study (*Todor et al., 2023*) and aligned using MUSCLE v5 (*Edgar, 2022*) with the 'Super5' algorithm. The phylogenetic tree was built using FastTree (*Price et al., 2010*) with the Jones–Taylor–Thornton (default) mode of amino acid evolution. All trees were visualized and edited in iTOL.

## Transposon mutagenesis

Transposon mutagenesis was performed by electroporation using the EZ-Tn5 <KAN-2> Tnp Transposome kit (Lucigen). Briefly, an overnight culture of *B. amyloliquefaciens* BK was diluted 100 fold in NCM medium (*Ito and Nagane, 2001*) and grown at 37°C on a rotary shaker (200 rpm) until an OD600 nm of 0.5. The cell culture was then supplemented with glycine, DL-threonine and Tween 80 at a final concentration of 3.89%, 1.06%, and 0.03%, respectively, and grown at 37°C on a rotary shaker (200 rpm) for 1 hr. The culture was then cooled on ice for 30 min and collected by centrifugation (4000 × $g$, 10 min) at 4°C. The cell pellet was washed three times with ETM (*Zhang et al., 2011*) buffer. The washed cell pellet was resuspended in 100 μl of ETM buffer containing 0.25 mM $KH_2PO_4$ and 0.25 mM $K_2HPO_4$ and 1 μl of the EZ-Tn5 transposome was added. The cell mixture was transferred to a 1-mm electroporation cuvette and pulsed using a Gene Pulser Xcell Electroporation System (Bio-Rad) using 2.1 kV/cm, 150 Ω, and 36 μF. The mixture was immediately and gently mixed with 1 ml NCM medium containing 0.38 M mannitol and recovered in a round-bottom tube (VWR) at 37°C on a rotary shaker (120 rpm) for 3 hr. After recovery, the cells were concentrated by centrifugation and spread on an LB plate with 50 μg/ml kanamycin and incubated at 37°C overnight and single colonies collected for subsequent analysis. A library containing 50 transposon mutants was obtained. In the mutants examined, each strain contained ≥4 transposon insertions.

## Screening of bacillus strains lacking anti-*N. gonorrhoeae* activity

The transposon mutants of *B. amyloliquefaciens* BK were grown overnight in LB medium at 30°C. Each overnight culture was then diluted 1:5000, and 1 μl of the diluted culture was spotted onto a GCB agar plate swabbed with *N. gonorrhoeae* cells. The plate was then incubated overnight at 37°C with 5% $CO_2$. The mutant strain (Tn5-3) lacking anti-*N. gonorrhoeae* activity was identified due to its failure to produce a zone of growth inhibition in the resulting *N. gonorrhoeae* lawn.

## Mutant development

For natural mutagenesis, single colonies of *N. gonorrhoeae* cells were swabbed on GCB agar plates and grown at 37°C with 5% $CO_2$ for 16 hr. The grown cells were collected by polyester swab tip (Puritan) in GCB medium and cell number was calculated by measuring optical density (OD, estimated value of OD 0.7 = 5 × $10^8$ CFU (colony-forming unit)/ml). Cells were then uniformly spread out on GCB agar plate containing 4× MIC of oxydifficidin (1 μg/ml for MS11 and 8 μg/ml for MS11 Δ*dedA*) and grown at 37°C with 5% $CO_2$ until mutant colonies are observed.

## Mutant construction

For *dedA* gene deletion, the 3′ (Primer: *dedA* 3′ overhang F/R) and 5′ (Primer: *dedA* 5′ overhang F/R) overhang regions of *N. gonorrhoeae dedA* gene and a kanamycin resistance cassette (Primer: kan$^R$ (kanamycin resistance) cassette F/R) were amplified using GoTaq master mix (Promega). The fragments were then assembled using NEBuilder HiFi DNA assembly master mix (NEB) to place the cassette sequence positioned in the middle. Transformation of the construct to *N. gonorrhoeae* was done by using spot transformation protocol (*Dillard, 2011*). For mutant *rplL*_R76C, the mutated *rplL* gene including the 3′ and 5′ overhangs (Primer: *rplL*_R76C F/R) was amplified using *N. gonorrhoeae* MS11 Δ*dedA rplL*_R76C as the PCR template. Another construct incorporating a kanamycin resistance cassette with the 3′ (Primer: *trpB-lga* 3′ overhang F/R) and 5′ (Primer: *trpB-lga* 5′ overhang F/R) overhangs from locus 272,353 bp (*trpB-lga*) of *N. gonorrhoeae* MS11 genome was assembled using HiFi assembly. These two constructs were co-transformed (*Dalia, 2018*) into *N. gonorrhoeae* MS11 cells. The sequence verification was carried out using Sanger sequencing services provided by Genewiz.

### Genomic sequencing and mutational analysis

Genomic DNA from *Neisseria* parent strains and resistance mutants was isolated using PureLink Microbiome DNA purification kit (Invitrogen) according to the manufacturer's instructions. The whole-genome sequencing was performed by a MiSeq Reagent Kit v3 using Illumina MiSeq system following the manufacturer's instructions. Genomes were assembled and mapped using corresponding reference genomes. The single-nucleotide polymorphism was detected by aligning the mutant's sequencing reads to the genomic sequence of the parent strains.

### Time-dependent killing assay

*N. gonorrhoeae* cells were grown on a GCB agar plate at 37°C with 5% $CO_2$ for 16 hr and the overnight cells were normalized to ~$10^5$ CFU/ml with GCB medium in a 14-ml round-bottom tube (VWR). The normalized cell culture was supplemented with 8× MIC of oxydifficidin (2 µg/ml for MS11 and 16 µg/ml for MS11 *ΔdedA*) and incubated at 37°C with 5% on a rotary shaker (200 rpm). 100 µl of the culture was collected and mixed thoroughly into 10 ml of GCB medium at 5, 10, and 30 min, respectively, then 200 µl of the diluted culture was spread on a GCB agar plate and grown at 37°C with 5% $CO_2$ overnight. Colonies were counted the following day. The killing assays were done in triplicate.

### Cell lysis assay

The cell lysis assay was conducted using SYTOX green (Invitrogen) following the manufacturer's instructions. Briefly, 1 ml of cell suspension in Live Cell Imaging Solution at OD 0.7 was stained by adding 0.4 µl of 5 mM SYTOX solution and incubated in the dark at RT for 5 min. Subsequently, 30 µl of the stained culture was added into a 384-well Flat Clear Bottom Black plate (Corning) and the fluorescence signal was recorded by an Infinite M NANO$^+$ (TECAN) with Excitation/Emission = 488/523 nm. The interval time was set to 7 s. Once the signal reached equilibrium, 30 µl of antibiotics diluted in Live Cell Imaging Solution at 16× MIC was added to the culture and mixed thoroughly. The fluorescence signal monitoring was then continued for 1 hr. The cell lysis assays were done in triplicate.

### Cell depolarization assay

The cell depolarization assay was conducted using DiSC$_3$(5) dye (Invitrogen) following the manufacturer's instructions. Briefly, 1 ml of cell suspension in Live Cell Imaging Solution at OD 0.7 was stained by adding 1 µl of 2 mM SYTOX solution and incubated in the dark at RT for 15 min. Then, 30 µl of the stained culture was added into a 384-well Flat Clear Bottom Black plate (Corning) and the fluorescence signal was recorded by an Infinite M NANO$^+$ (TECAN) with Excitation/Emission = 622/675 nm. The interval time was set to 7 s. Once the signal reached equilibrium, 30 µl of antibiotics diluted in Live Cell Imaging Solution at 16× MIC was added to the culture and mixed thoroughly. The fluorescence signal monitoring was then continued for 1 hr. The cell depolarization assays were done in triplicate.

### Drug accumulation assay

*N. gonorrhoeae* cells were grown on a GCB agar plate at 37°C with 5% $CO_2$ for 9 hr and the overnight cells were normalized to ~$10^8$ CFU/ml with GCB medium in a 14-ml round-bottom tube (VWR). Cells were incubated with 0.125 µg/ml of oxydifficidin at 37°C with 5% $CO_2$ on a rotary shaker (200 rpm) for 3 hr. Cell pellets were collected by centrifugation at RT, washed twice with fresh GCB medium, and resuspended in an equal volume of GCB medium to that of the supernatant. The cell suspension and supernatant were then separately extracted by ethyl acetate in a 1:1 ratio. The extracts were evaporated to dryness under vacuum, resuspended in 100 µl of methanol and then 3 µl of the resuspension was injected to LC–HRMS for the quantification of oxydifficidin using target peak area. Experiment 1 (oxydifficidin and tetracycline) was done in duplicate, and experiment 2 (oxydifficidin and chloramphenicol) was done in triplicate.

### In vitro transcription and translation inhibition assays

The in vitro transcription assay was conducted using the HiScribe T7 High Yield RNA Synthesis Kit (NEB) following the manufacturer's instructions. 24 µM of oxydifficidin was added to the transcription mixture, and 20 mM of EDTA was added to provide a positive control for inhibition. The yielded RNA was detected by a BioAnalyzer (Agilent). The in vitro translation assay was performed using PURExpress In Vitro Protein Synthesis Kit (NEB). 24 µM of antibiotics were added individually to the

translation mixture and incubated at 37°C for 4 hr, the samples were then analyzed by SDS–PAGE in a 5–20% gradient gel.

## Membrane integrity assay

The membrane integrity assay was conducted using BacLight Bacterial Viability Kit (Invitrogen) following the manufacturer's instructions. Briefly, 1 ml of cell suspension at OD 0.7 was stained by adding 3 µl of the dye mixture and incubated in the dark at RT for 15 min. Subsequently, 1 µl of the stained culture was added onto the center of a round cover glass (Warner Instruments), a 1 mm × 1 mm GCB agar pad was overlayed on the culture. The sample was then imaged by an Eclipse Ti Microscope with a 60× objective (Nikon). DIC, GFP, and Texas Red fluorescence channels were applied to the imaging. ImageJ was used to analyze all images. Membrane integrity was assessed by calculating the ratio of the number of green cells to red cells. Boiled cells were served as the negative control. The killing assays were done in 10 replicates.

## Micro-colony formation assay

100 µl of cell suspension at OD 0.7 were added to 1 ml of fresh GCB medium in a 6-well plate (VWR) and incubated at 37°C with 5% $CO_2$ for 3 hr. Pictures were then taken using a 20× objective (Nikon).

## SEM of *N. gonorrhoeae*

30 µl of cell suspension at OD 0.7 was spotted onto 12 mm glass or Aclar film round coverslips coated with 0.1% Poly-lysin. A drop of fixative (100 µl of 2% glutaraldehyde, 4% formaldehyde in 0.1 M sodium cacodylate buffer pH 7.2) was added on top of the coverslip for 30 min. Additional fixative was then added to the dish containing the coverslips and left it in the fridge overnight. Samples were gently washed three times with buffer for 5–10 min each time and postfixed with osmium tetroxide 1% in 0.1 M sodium cacodylate buffer pH 7.2 for 1 hr at RT. After rinsing three times with buffer, samples were dehydrated in a graded series of ethanol concentrations 30%, 50%, 70%, and 90% for 10 min each and three times in 100% ethanol with molecular sieves for 15 min each and critical point dried in a Tousimis Autosamdri 931. Samples were coated with 10 nm of iridium using a Leica ACE600 sputter coater. Imaging was done in a JEOL JSM-IT500HR at 5.0 kV.

## Statistical analysis

Statistical analysis was performed using Prism 10 (GraphPad). Group data are presented as means with SEM. The significance was determined using one-way ANOVA (and nonparametric or mixed). p values less than 0.05 were considered significant.

# Acknowledgements

We thank Nathan Weyand at Ohio University for providing *Neisseria* stains, the Bacillus Genetic Stock Center for providing *Bacillus amyloliquefaciens* strains, Rinat Abzalimov and James Aramini at Advanced Science Research Center (ASRC) for performing initial LC–MS and NMR, David Dubnau and Jeanette Hahn at Rutgers New Jersey Medical School for assistance with Bacillus genetics, and Hilda Amalia Pasolli at The Rockefeller University Electron Microscopy Resource Center for collecting scanning electron microscope images. We acknowledge the members of the MMBL lab at Brooklyn College for their help and input in this work. This work was supported by 5R35GM122559 (SFB), NIH AI116566 (NB), and iBio Initiative (NB).

# Additional information

## Funding

| Funder | Grant reference number | Author |
| --- | --- | --- |
| National Institute for Health Care Management Foundation | AI116566 | Nicolas Biais |

| Funder | Grant reference number | Author |
|---|---|---|
| National Institute for Health Care Management Foundation | 5R35GM122559 | Sean F Brady |
| iBio Initiative | | Nicolas Biais |

The funders had no role in study design, data collection, and interpretation, or the decision to submit the work for publication.

## Author contributions

Jingbo Kan, Conceptualization, Data curation, Formal analysis, Validation, Investigation, Visualization, Methodology, Writing – original draft, Project administration, Writing – review and editing; Adrian Morales-Amador, Data curation, Formal analysis, Validation, Investigation, Visualization, Methodology; Yozen Hernandez, Data curation, Software, Formal analysis, Methodology; Melinda A Ternei, Logan W MacIntyre, Formal analysis, Methodology; Christophe Lemetre, Software, Formal analysis, Methodology; Nicolas Biais, Sean F Brady, Conceptualization, Resources, Data curation, Formal analysis, Supervision, Funding acquisition, Validation, Investigation, Visualization, Methodology, Writing – original draft, Project administration, Writing – review and editing

## Author ORCIDs

Jingbo Kan  https://orcid.org/0009-0006-1917-6577
Adrian Morales-Amador  https://orcid.org/0000-0001-7944-308X
Yozen Hernandez  https://orcid.org/0000-0003-3349-8856
Nicolas Biais  https://orcid.org/0009-0004-6054-594X
Sean F Brady  https://orcid.org/0000-0001-5967-8586

Reviewer #1 (Public review): https://doi.org/10.7554/eLife.99281.3.sa1
Reviewer #2 (Public review): https://doi.org/10.7554/eLife.99281.3.sa2
Reviewer #3 (Public review): https://doi.org/10.7554/eLife.99281.3.sa3
Author response https://doi.org/10.7554/eLife.99281.3.sa4

# Additional files

## Supplementary files

Supplementary file 1. Activity of oxydifficidin against of *Neisseria* mutants.
Supplementary file 2. $^1$H- and $^{13}$C-NMR data of oxydifficidin (800 MHz, 298 K, CD$_3$OD–D$_2$O (1:1)).
Supplementary file 3. Primer sequences used in this study.
MDAR checklist

## Data availability

The assembled sequencing data of *Bacillus amyloliquefaciens* have been submitted to the NCBI genome database under accession no. GCA_019093835.1.

The following dataset was generated:

| Author(s) | Year | Dataset title | Dataset URL | Database and Identifier |
|---|---|---|---|---|
| Kan J | 2021 | Genome assembly ASM1909383v1 | https://www.ncbi.nlm.nih.gov/datasets/genome/GCF_019093835.1/ | NCBI Assembly, GCF_019093835.1 |

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

## Appendix 1

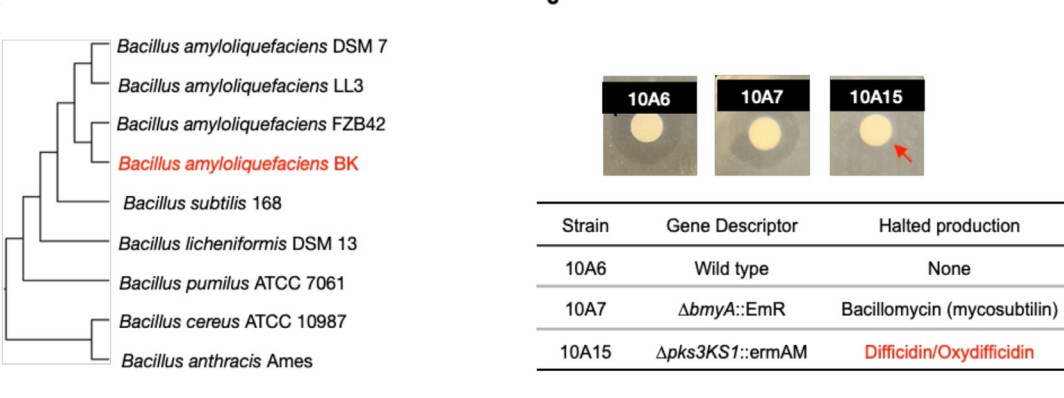

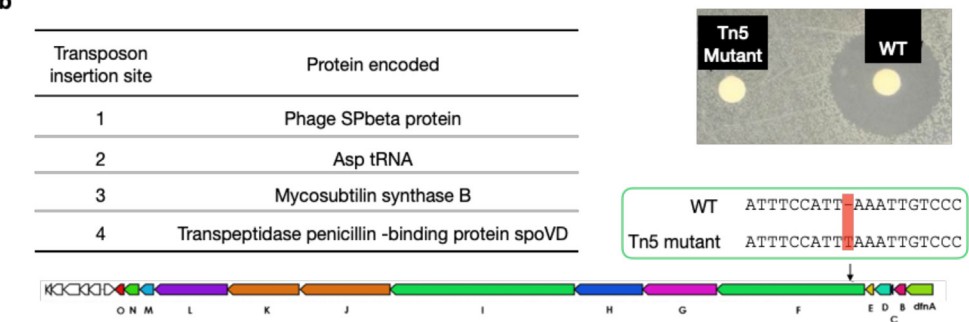

**Appendix 1—figure 1.** Mutagenesis results of oxydifficidin-producing *Bacillus* spp. (**a**) Genome-based phylogenetic tree containing *Bacillus amyloliquefaciens* BK and closely related *Bacillus* spp. The tree was built by Genome Clustering of MicroScope using neighbor-joining method. The NCBI accession numbers of *Bacillus* strains used in the tree are GCA_000196735.1, GCA_000204275.1, GCA_000015785.2, GCA_019093835.1, GCA_000009045.1, GCA_000011645.1, GCA_000172815.1, GCA_000008005.1, and GCA_000007845.1 (from top to bottom). (**b**) Disc diffusion assay of a methanol extract from cultures of WT *B. amyloliquefaciens* BK (WT) and a Tn5 mutant. The test lawn was *N. gonorrhoeae*. The table shows all transposon insertion sites in the Tn5 mutant strain. The Tn5 strain also contains a frame-shift mutation in the *difF* gene; red box highlights the location of frame-shift mutation in the oxydifficidin biosynthetic gene cluster (BGC). (**c**) Disc diffusion assay of a methanol extract from cultures of WT and BGC knockout strains of *Bacillus amyloliquefaciens* FZB42. The test lawn was *N. gonorrhoeae*. Strain genotypes are shown in the table. Red arrow indicates that only strain 10A15 no longer produce the anti-*N. gonorrhoeae* compound.

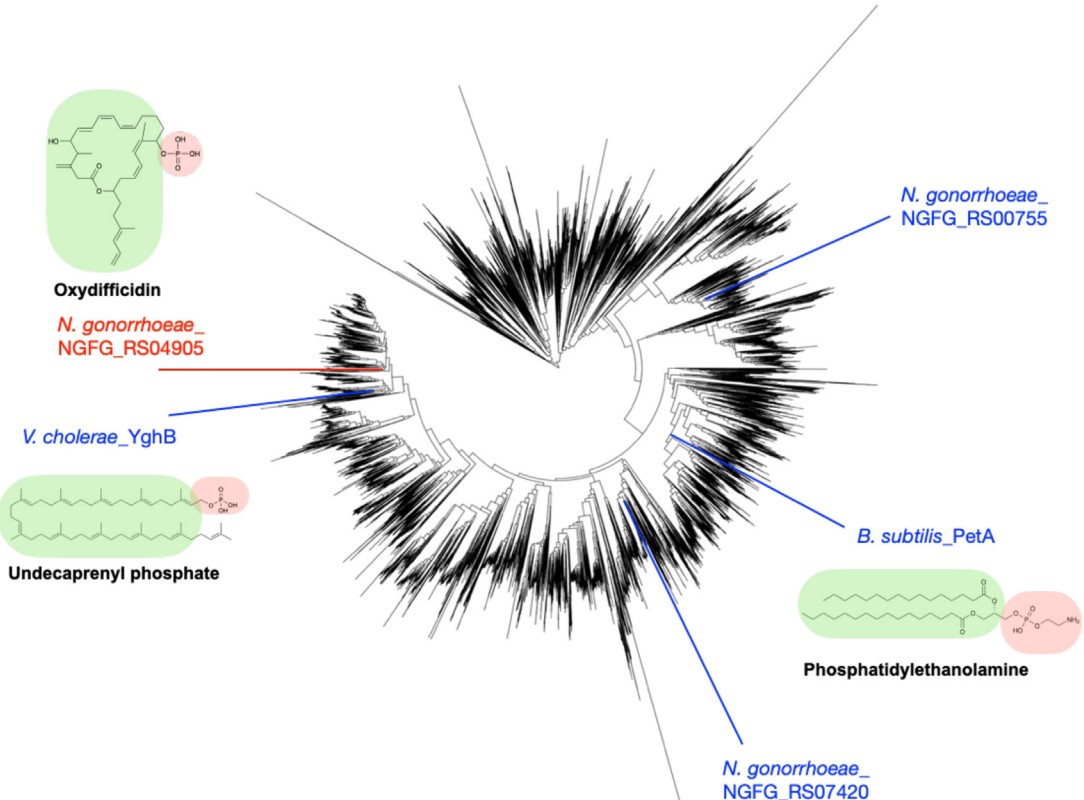

**Appendix 1—figure 2.** Phylogenetic tree of 15,825 bacterial DedA family proteins. The tree was built by MUSCLE v5 and FastTree and visualized using iTOL. *N. gonorrhoeae* NGFG_RS04905 highlighted in red represents the DedA gene associated with the activity of oxydifficidin. *N. gonorrhoeae* NGFG_RS07420 and *N. gonorrhoeae* NGFG_RS00755 represents two other DedA family proteins in *N. gonorrhoeae.*

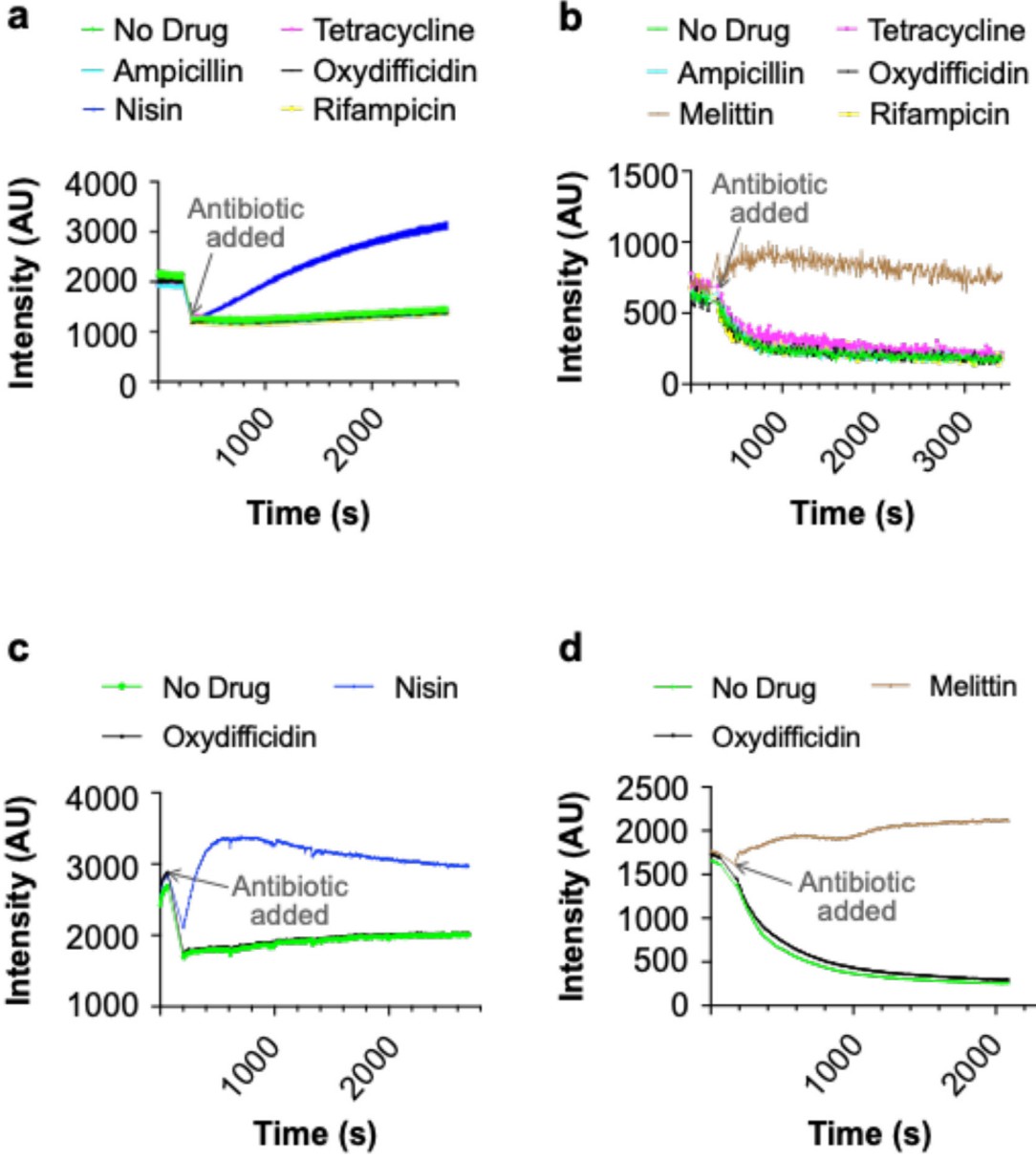

**Appendix 1—figure 3.** Oxydifficidin does not lyse or depolarize the membrane of *N. gonorrhoeae*. (**a**) Lysis assay using SYTOX green dye and 8× the minimum inhibitory concentration (MIC) of each antibiotic. (**b**) Depolarization assay using $DiSC_3(5)$ dye and 8× the MIC of each antibiotic. (**c**) Lysis assay using SYTOX green dye with 100× the MIC of oxydifficidin and 32× the MIC of nisin. (**d**) Depolarization assay using $DiSC_3(5)$ dye with 100× the MIC of oxydifficidin and the 32× the MIC of melittin (*n* = 3 for all assays).

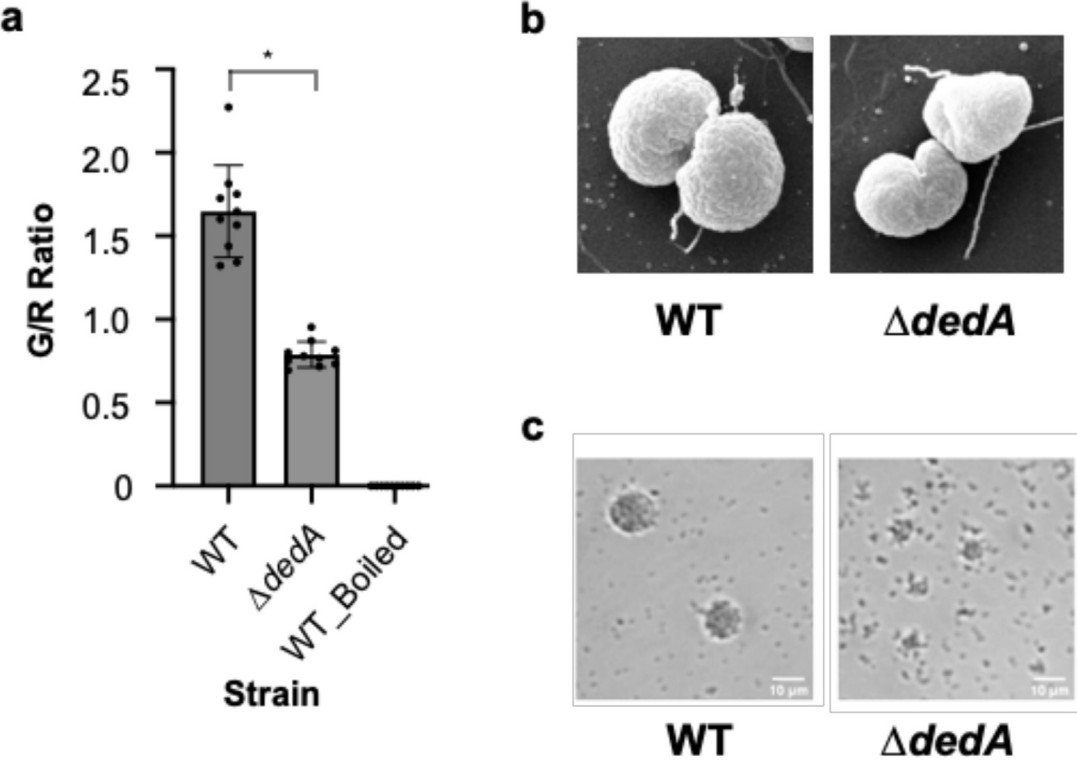

**Appendix 1—figure 4.** Mutations in *dedA* affect cell morphology and pili functionality of *N. gonorrhoeae*. (**a**) Membrane integrity assay of *N. gonorrhoeae* WT and *dedA* deletion mutant (*ΔdedA*) cells using SYTO 9 and propidium iodide. Cell integrity was assessed using the ratio of green- to red-stained cell count. *p < 0.05. (**b**) Scanning electron microscope pictures of *N. gonorrhoeae* WT and *ΔdedA* cells. (**c**) Micro-colony formation assay of *N. gonorrhoeae* WT and *ΔdedA* cells.

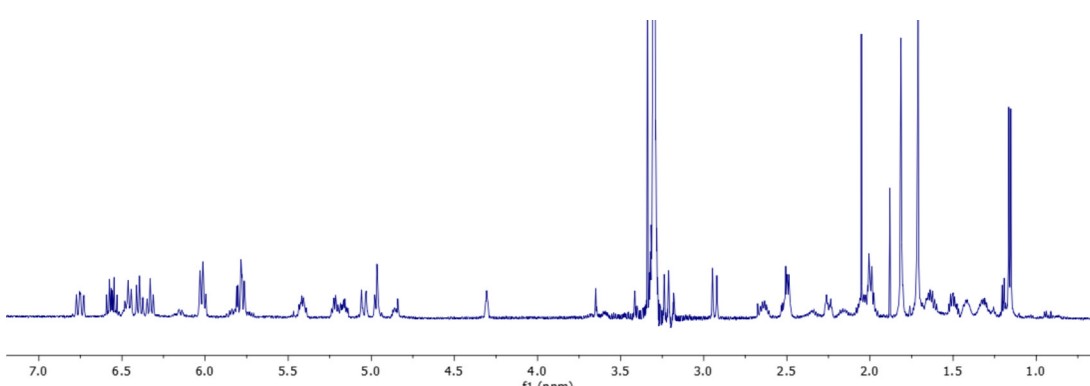

**Appendix 1—figure 5.** $^1$H-NMR spectrum of oxydifficidin (800 MHz, 298 K, CD$_3$OD–D$_2$O (1:1)).

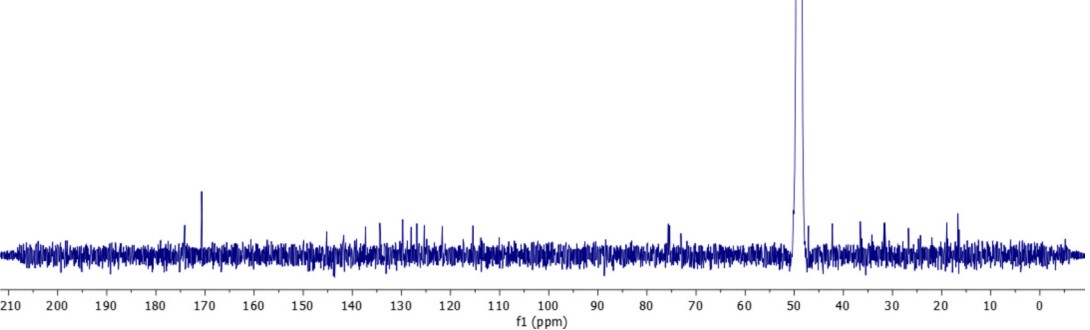

**Appendix 1—figure 6.** $^{13}$C-NMR spectrum of oxydifficidin (800 MHz, 298 K, CD$_3$OD–D$_2$O (1:1)).

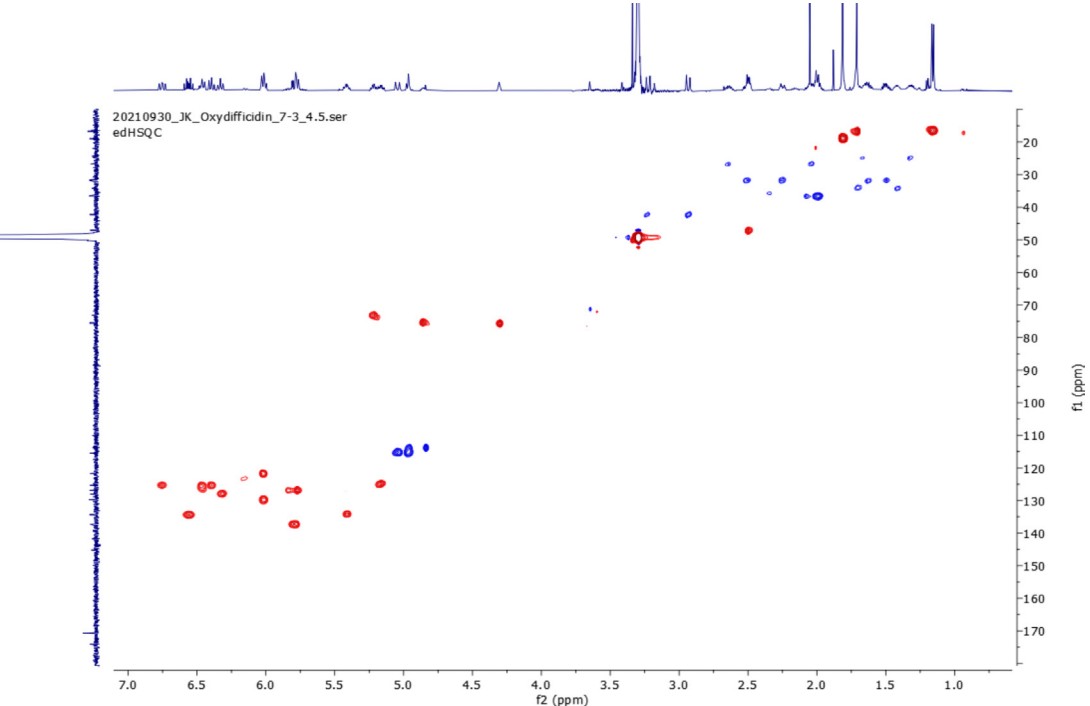

**Appendix 1—figure 7.** edHSQC spectrum (800 MHz, 298 K, CD$_3$OD–D$_2$O (1:1)) of oxydifficidin.

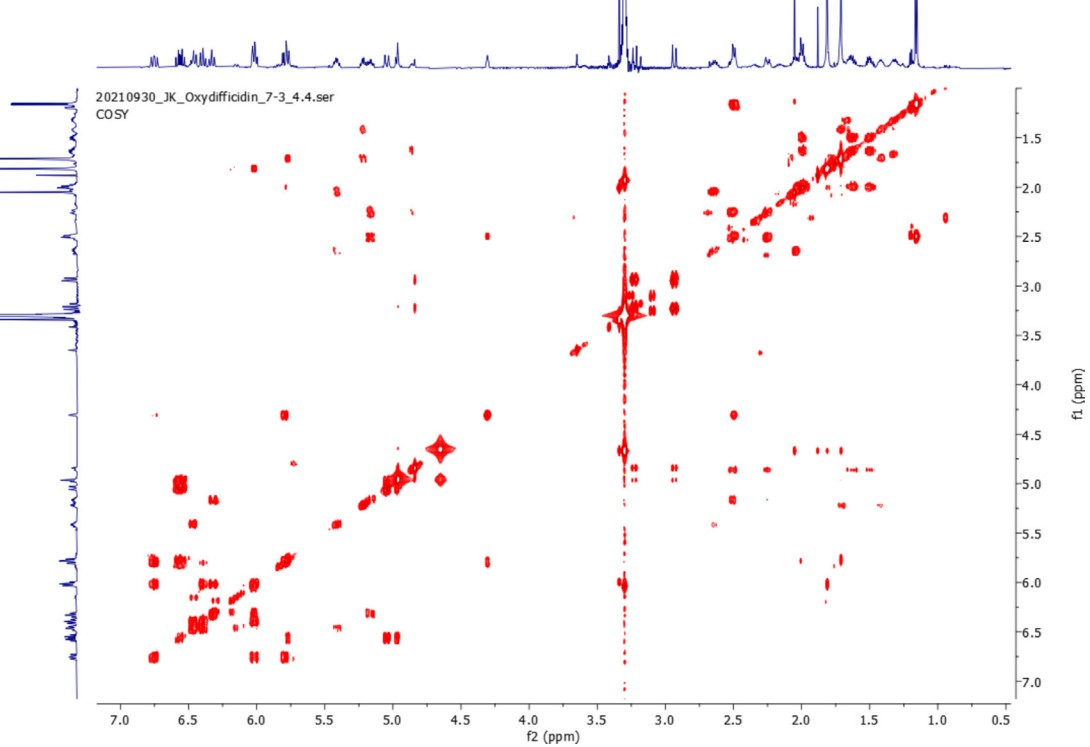

**Appendix 1—figure 8.** COSY spectrum (800 MHz, 298 K, CD$_3$OD–D$_2$O (1:1)) of oxydifficidin.

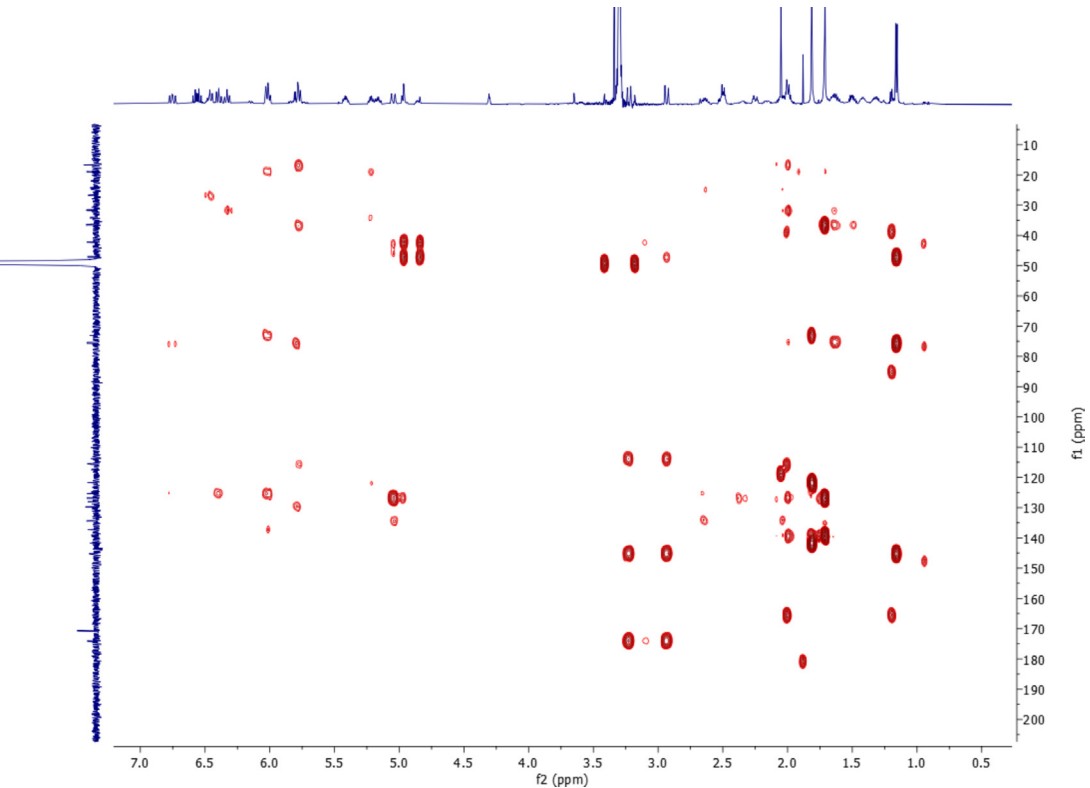

**Appendix 1—figure 9.** $^1$H–$^{13}$C HMBC spectrum (800 MHz, 298 K, CD$_3$OD–D$_2$O (1:1)) of oxydifficidin.

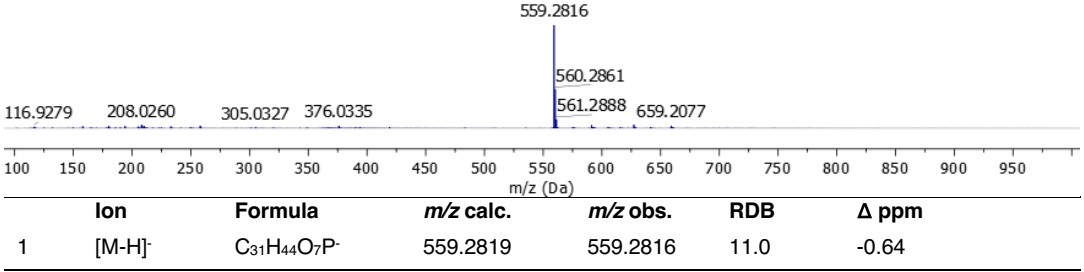

**Appendix 1—figure 10.** Partial COSY (left) and ROESY (right) comparison and key ROESY correlations of oxydifficidin (800 MHz, 298 K, CD₃OD–D₂O (1:1)).

| | Ion | Formula | *m/z* calc. | *m/z* obs. | RDB | Δ ppm |
|---|---|---|---|---|---|---|
| 1 | [M-H]⁻ | $C_{31}H_{44}O_7P^-$ | 559.2819 | 559.2816 | 11.0 | -0.64 |

**Appendix 1—figure 11.** Full HRMS and annotation of oxydifficidin [M–H]⁻ parental ion.

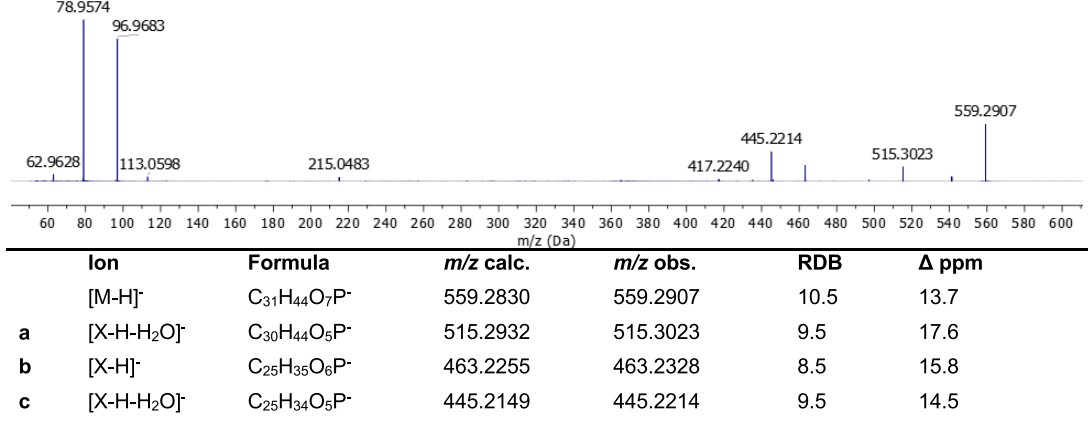

| | Ion | Formula | *m/z* calc. | *m/z* obs. | RDB | Δ ppm |
|---|---|---|---|---|---|---|
| | [M-H]⁻ | C₃₁H₄₄O₇P⁻ | 559.2830 | 559.2907 | 10.5 | 13.7 |
| a | [X-H-H₂O]⁻ | C₃₀H₄₄O₅P⁻ | 515.2932 | 515.3023 | 9.5 | 17.6 |
| b | [X-H]⁻ | C₂₅H₃₅O₆P⁻ | 463.2255 | 463.2328 | 8.5 | 15.8 |
| c | [X-H-H₂O]⁻ | C₂₅H₃₄O₅P⁻ | 445.2149 | 445.2214 | 9.5 | 14.5 |
| d | [X-H]⁻ | H₂O₄P⁻ | 96.9696 | 96.9683 | 0.5 | 13.9 |
| e | [X-H]⁻ | O₃P⁻ | 78.9591 | 78.9574 | 0.5 | -21.5 |

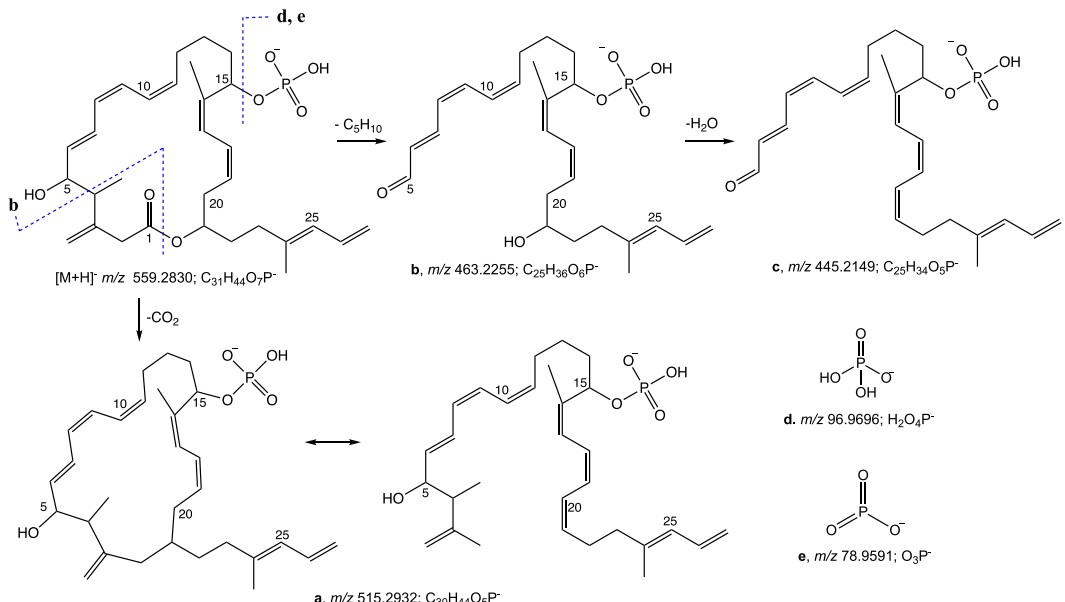

**Appendix 1—figure 12.** ESI MS/MS spectrum and fragment annotation of oxydifficidin [M–H]⁻ ion.

