## [Editor Report · eLife Assessment]

Kan et al. report the discovery of a Bacillus amyloliquifaciens strain that kills Nerisseria gonorrhoeae via oxydifficidin which targets ribosomal proteins. Resistance occurred via mutation in the DedA flippase to influence oxydifficidin uptake. The overall mechanism of action is well described making this an **important** study with implications for combating clinical antibiotic resistance. The evidence presented is **convincing** due to rigour employed in the methodological approach. The authors should consider performing a more comprehensive genetic analyses of DedA and RpIL in this clinically relevant strain. This work will be of broad interest to microbiologists and synthetic biologists.

---

## [Referee Report · Reviewer #1 (Public review)]

Summary:

Kan et al. report the serendipitous discovery of a Bacillus amyloliquefaciens strain that kills N. gonorrhoeae. They use TnSeq to identify that the anti-gonococcal agent is oxydifficidin and show that it acts at the ribosome and that one of the dedA gene products in N. gonorrhoeae MS11 is important for moving the oxydifficidin across the membrane.

Strengths:

- This is an impressive amount of work, moving from a serendipitous observation through TnSeq to characterize the mechanism by which Oxydifficidin works.

Weaknesses:

- The genetic diversity of dedA and rplL in N. gonorrhoeae is still not clear, as the authors looked at diversity of these genes in only 220 isolates (of unclear relationship to each other).

It's not so much a weakness as a source of confusion: how did the authors choose to screen a tiny transposon library of 50 mutants? Since they were surprised to find 4 transposon insertions (if I'm reading it correctly), what was the motivation for even looking at this small library? And since the mutation that led them to the biosynthetic gene cluster wasn't even a transposon insertion but a frameshift, it seems they had another huge episode of serendipity.

---

## [Referee Report · Reviewer #2 (Public review)]

Summary:

Kan et al. presents the discovery of oxydifficidin as a potential antimicrobial against N. gonorrhoeae, including multi-drug resistant strains. The authors show the role of DedA flippase assisted uptake and the specificity of RplL in the mechanism of action for oxydifficidin. This mode of action could potentially offer a new therapeutic avenue, providing a critical addition to the limited arsenal of antibiotics effective against gonorrhea.

Strengths:

This study shows the potential of revisiting anti-bacterial agents/products for antibacterial activity against modern-day-concerning pathogens and highlights a new anti-gonoccoal mechanism of action. Indeed there is a recent growing body of research to revisit potential antimicrobial agents and metabolites from cultured bacterial species. The discovery of oxydifficidin interaction with RplL and its DedA-assisted uptake mechanism opens new research directions in understanding and combating antibiotic resistant N. gonorrhoeae. The antimicrobial activity of oxydifficidin is also active against N. meningitidis, a closely related species. Methodologically, the study is rigorous employing various experimental techniques including Tn-mutagenesis (TraDIS, Tn-Seq).

Weaknesses:

While the study demonstrates the in vitro effectiveness of oxydifficidin, there is a lack of in vivo validation (i.e., animal models) for assessing pre-clinical potential of oxydifficidin. However, I acknowledge that this would be a tremendous amount of work and likely outside the scope of this study. Potential SNPs within dedA or RplL raises concerns about how quickly resistance could emerge in clinical settings.

---

## [Referee Report · Reviewer #3 (Public review)]

Summary:

The authors have shown that oxydifficidin is a potent inhibitor of Neisseria gonorrhoeae. They were able to identify the target of action to rpsL and showed that resistance could occur via mutation in the DedA flippase and RpsL.

Strengths:

This was a very thorough and clearly argued set of experiments that supported their conclusions.

Weaknesses:

There was no obvious weakness in the experimental design. Although it is promising that the DedA mutations resulted in attenuation of fitness, it remains an open question whether secondary rounds of mutation could overcome this selective disadvantage which was untried in this study.

Comments on revisions:

All of my suggestions were considered and the responses to the other reviewer's appears sound and has improved the manuscript.

---

## [Author Response]

The following is the authors’ response to the original reviews.

**eLife Assessment**
This useful study reports on the discovery of an antimicrobial agent that kills Neisseria gonorrhoeae. Sensitivity is attributed to a combination of DedA assisted uptake of oxydifficidin into the cytoplasm and the presence of a oxydifficidin-sensitive RplL ribosomal protein. Due to the narrow scope, the broader antibacterial spectrum remains unclear and therefore the evidence supporting the conclusions is incomplete with key methods and data lacking. This work will be of interest to microbiologists and synthetic biologists.

General comment about narrow scope: The broader antibacterial spectrum of oxydifficidin has been reported previously (S B Zimmerman et al., 1987). The main focus of this study is on its previously unreported potent anti-gonococcal activity and mode of action. While it is true that broad-spectrum antibiotics have historically played a role in effectively controlling a wide range of infections, we and others believe that narrow-spectrum antibiotics have an overlooked importance in addressing bacterial infections. Their advantage lies in their ability to target specific pathogens without markedly disrupting the human microbiota.

**Public Reviews:**

**Reviewer #1 (Public Review):**
Summary:Kan et al. report the serendipitous discovery of a Bacillus amyloliquefaciens strain that kills N. gonorrhoeae. They use TnSeq to identify that the anti-gonococcal agent is oxydifficidin and show that it acts at the ribosome and that one of the dedA gene products in N. gonorrhoeae MS11 is important for moving the oxydifficidin across the membrane.Strengths:This is an impressive amount of work, moving from a serendipitous observation through TnSeq to characterize the mechanism by which Oxydifficidin works.Weaknesses:(1) There are important gaps in the manuscript's methods.

The requested additions to the method describing bacterial sequencing and anti-gonococcal activity screening will be made. However, we do not think the absence of these generic methods reduces the significance of our findings.

(2) The work should evaluate antibiotics relevant to N. gonorrhoeae.

(1) It is not clear to us why reevaluating the activity of well characterized antibiotics against known gonorrhoeae clinical strains would add value to this manuscript. The activity of clinically relevant antibiotics against antibiotic-resistant *N. gonorrhoeae* clinical isolates is well described in the literature. Our use of antibiotics in this study was intended to aid in the identification of oxydifficidin’s mode of action. This is true for both Tables 1 and 2.

(2) If the reviewer insists, we would be happy to include MIC data for the following clinically relevant antibiotics: ceftriaxone (cephalosporin/beta-lactam), gentamicin (aminoglycoside), azithromycin (macrolide), and ciprofloxacin (fluoroquinolone).

(3) The genetic diversity of dedA and rplL in N. gonorrhoeae is not clear, neither is it clear whether oxydifficidin is active against more relevant strains and species than tested so far.

(1) We thank the reviewer for this suggestion. We aligned the DedA sequence from strain MS11 with DedA proteins from 220 *N. gonorrhoeae* strains that have high-quality assemblies in NCBI. The result showed that there are no amino acid changes in this protein. Using the same method, we observed several single amino acid changes in RplL. This included changes at A64, G25 and S82 in 4 strains with one change per strain. These sites differ from R76 and K84, where we identified changes that provide resistance to oxydifficidin. Notably, in a similar search of representative *Escherichia*, *Chlamydia*, *Vibrio*, and *Pseudomonas* NCBI deposited genomes, we did not identify changes in RplL at position R76 or K84.

(2) While the usefulness of screening more clinically relevant antibiotics against clinical isolates as suggested in comment 2 was not clear to us, we agree that screening these strains for oxydifficidin activity would be beneficial. We have ordered *Neisseria gonorrhoeae* strain AR1280, AR1281 (CDC), and *Neisseria meningitidis* ATCC 13090. They will be tested when they arrive.

**Reviewer #2 (Public Review):**
Summary:Kan et al. present the discovery of oxydifficidin as a potential antimicrobial against N. gonorrhoeae, including multi-drug resistant strains. The authors show the role of DedA flippase-assisted uptake and the specificity of RplL in the mechanism of action for oxydifficidin. This novel mode of action could potentially offer a new therapeutic avenue, providing a critical addition to the limited arsenal of antibiotics effective against gonorrhea.Strengths:This study underscores the potential of revisiting natural products for antibiotic discovery of modern-day-concerning pathogens and highlights a new target mechanism that could inform future drug development. Indeed there is a recent growing body of research utilizing AI and predictive computational informatics to revisit potential antimicrobial agents and metabolites from cultured bacterial species. The discovery of oxydifficidin interaction with RplL and its DedA-assisted uptake mechanism opens new research directions in understanding and combating antibiotic-resistant N. gonorrhoeae. Methodologically, the study is rigorous employing various experimental techniques such as genome sequencing, bioassay-guided fractionation, LCMS, NMR, and Tn-mutagenesis.Weaknesses:The scope is somewhat narrow, focusing primarily on N. gonorrhoeae. This limits the generalizability of the findings and leaves questions about its broader antibacterial spectrum. Moreover, while the study demonstrates the in vitro effectiveness of oxydifficidin, there is a lack of in vivo validation (i.e., animal models) for assessing pre-clinical potential of oxydifficidin. Potential SNPs within dedA or RplL raise concerns about how quickly resistance could emerge in clinical settings.

(1) Spectrum/narrow scope: The broader antibacterial spectrum of oxydifficidin has been reported previously (S B Zimmerman et al., 1987). The focus of this study is on its previously unreported potent anti-gonococcal activity and its mode of action. While it is true that broad-spectrum antibiotics have historically played a role in effectively controlling a wide range of infections, we and others believe that narrow-spectrum antibiotics have an overlooked importance in addressing bacterial infections. Their advantage lies in their ability to target specific pathogens without markedly disrupting the human microbiota.

(2) Animal models: We acknowledge the reviewer’s insight regarding the importance of in vivo validation to enhance oxydifficidin’s pre-clinical potential. However, due to the labor-intensive process needed to isolate oxydifficidin, obtaining a sufficient quantity for animal studies is beyond the scope of this study. Our future work will focus on optimizing the yield of oxydifficidin and developing a topical mouse model for subsequent investigations.

(3) Potential SNPs: Please see our response to Reviewer #1’s comment 3. We acknowledge that potential SNPs within dedA and rplL raise concerns regarding clinical resistance, which is a common issue for protein-targeting antibiotics. Yet, as pointed out in the manuscript, obtaining mutants in the lab was a very low yield endeavor.

**Reviewer #3 (Public Review):**
Summary:The authors have shown that oxydifficidin is a potent inhibitor of Neisseria gonorrhoeae. They were able to identify the target of action to rplL and showed that resistance could occur via mutation in the DedA flippase and RplL.Strengths:This was a very thorough and clearly argued set of experiments that supported their conclusions.Weaknesses:There was no obvious weakness in the experimental design. Although it is promising that the DedA mutations resulted in attenuation of fitness, it remains an open question whether secondary rounds of mutation could overcome this selective disadvantage which was untried in this study.

We thank the reviewer for the positive comment. We agree that investigating factors that could compensate for the fitness attenuation caused by DedA mutation would enhance our understanding of the role of DedA.

**Recommendations for the authors:**

**Reviewer #1 (Recommendations For The Authors):**
(1) The use of the term "N. gonorrhoeae wildtype" should not be used. It is uninformative, as the species contains a large amount of diversity. Instead, please name the strain. From Figure 1, it looks like the authors used MS11. Since MS11 is a longstanding lab strain and likely does not reflect circulating N. gonorrhoeae, and since H041 is no longer in circulation, the authors should ideally test the compound against more representative strains of N. gonorrhoeae. This includes panels of isolates available through the CDC, for example (https://www.cdc.gov/drugresistance/resistance-bank/index.html). I encourage the authors to include FC428 or another recently identified isolate with the penA 60 allele to demonstrate oxydifficidin's activity against contemporary concerning isolates/lineages.

(1) “*N. gonorrhoeae* MS11” is now used instead of “*N. gonorrhoeae* WT” in this manuscript.

(2) In our revised manuscript, we have added MIC data for recently identified *Neisseria gonorrhoeae* isolates AR#1280 and AR#1281 which contain the penA 60 allele (Table 1). The data shows oxydifficidin maintains its potent activity against these multidrug-resistant strains. We also added a description of this data to the results section as shown below.

Original text: “Oxydifficidin was more potent against *N. gonorrhoeae* MS11 than almost all other antibiotics we tested. In fact, it was only slightly less active than the highly optimized third-generation cephalosporin, ceftazidime.([18]) However, unlike third-generation cephalosporins, oxydifficidin retained activity against the multidrug resistant H041 clinical isolate (Table 1).([4]) H041 is resistant to the “standard of care” cephalosporin ceftriaxone (2 µg/mL) as well as a number of other antibiotics that are normally active against *N. gonorrhoeae* (penicillin G, 4 µg/mL; cefixime, 8 µg/mL; levofloxacin, 32 µg/mL).”

Changed to: “Oxydifficidin was more potent against *N. gonorrhoeae* MS11 than most other antibiotics we tested. Notably, unlike clinically used antibiotics such as ceftriaxone, azithromycin, and ciprofloxacin, oxydifficidin retained activity against all multidrug-resistant clinical isolates we examined (Table 1).” (Line 77-79)

(2) Does oxydifficidin have activity against N. meningitidis? It is the species most closely related to N. gonorrhoeae and the other pathogenic Neisseria.

Oxydifficidin has potent activity against *N. meningitidis* ATCC 13090. In our revised manuscript, we have included its MIC data in Figure 1c.

(3) Given claims that oxydifficidin activity in N. gonorrhoeae as compared to other Neisseria reflects N. gonorrhoeae's dedA and sensitive rplL, it would be good to assess the allelic diversity of these genes in N. gonorrhoeae. There are over 20,000 genomes from clinical isolates of N. gonorrhoeae in databases. It should be straightforward to check whether dedA and rplL allelic variants already exist in the population. Should variants be observed, oxydifficidin should be tested against the associated strains of N. gonorrhoeae.

Response: We thank the reviewer for this suggestion. We aligned the DedA sequence from strain MS11 with DedA proteins from 220 *N. gonorrhoeae* strains that have high-quality assemblies in NCBI. The result showed that there are no amino acid changes in this protein. Using the same method, we observed several single amino acid changes in RplL. This included changes at A64, G25 and S82 in 4 strains with one change per strain. These sites differ from R76 and K84, where we identified changes that provide resistance to oxydifficidin. Notably, in a similar search of representative *Escherichia*, *Chlamydia*, *Vibrio*, and *Pseudomonas* NCBI deposited genomes, we did not identify changes in RplL at position R76 or K84.

New text: “A survey of 220 *N. gonorrhoeae* strains with high-quality assemblies in NCBI found no mutations in the DedA protein.” (Line 104-105)

“These two mutations were not found in the survey of the same collection of *N. gonorrhoeae* strains used to look for DedA mutations.” (Line 143-144)

(4) Clinically relevant antibiotics for N. gonorrhoeae are penicillin, tetracycline, spectinomycin, gentamicin, ciprofloxacin, azithromycin, ceftriaxone; moreover, zoliflodacin and gepotidacin have reportedly successfully completed phase 3 trials. The authors should redo their MIC testing with these antibiotics (e.g., for Figures 1 and 2 and Tables 1 and 2), both because this will enable direct comparison with the many clinical isolates that have undergone testing and because these are the drugs most pertinent to clinical practice. Ampicillin, ceftazidime, chloramphenicol, bacitracin, and daptomycin are not relevant. Could the authors explain why they tested vancomycin, polymyxin B, irgasan, melittin, avilamycin, and thiostrepton?

Our use of antibiotics with diverse modes of action (e.g. vancomycin, polymyxin B, irgasan, melittin, avilamycin, and thiostrepton) in this study was intended to aid in the identification of oxydifficidin’s mode of action. This is true for both Tables 1 and 2.

To address the reviewer’s concern, in our revised manuscript, we have added MIC data for the following clinically relevant antibiotics: ceftriaxone (cephalosporin/beta-lactam), gentamicin (aminoglycoside), azithromycin (macrolide), and ciprofloxacin (fluoroquinolone) to Table 1.

(5) Please describe the characteristics of the transposon library (finding four transposons in a single strain does seem unexpected, given how most transposon libraries aim for one transposon insertion per strain).

We understand that one transposon insertion per strain is ideal for transposon libraries. This *Bacillus* strain proved to be recalcitrant to genetic manipulation. In the rare cases where we obtained resistance colonies upon electroporation with the transposon, all colonies contained multiple (≥ 4) transposon insertions. This made it impractical to build a library with one transposon insertion per library member.

We assumed that the anti-*N. gonorrhoeae* activity most likely originated from a natural product BGC, which typically range from 10-100 kb in size.

Based on the average of 50 kb per BGC, ~80 transposon insertions would be required to fully search the 4.2 Mb genome of *Bacillus amyloliquefaciens* BK for a BGC. At 4 mutations per transformant, 1x coverage of the genome would require only 20 library members.

After extensive electroporation of transposon into *Bacillus amyloliquefaciens* BK, we were able to obtain a library of 50 members, including one mutant (Tn5-3) that lacked anti-*N. gonorrhoeae* activity.

New text added to the methods section:

“A library containing 50 transposon mutants was obtained. In the mutants examined, each strain contained ≥4 transposon insertions” (Line 337-339)

(6) Please describe in the methods how you sequenced and annotated the genome of Bacillus amyloliquefaciens BK.

The sequencing method is now described in “Genomic Sequencing and annotation of *Bacillus amyloliquefaciens*” section. The genome of *Bacillus amyloliquefaciens* BK was not fully annotated. Mutations were identified as described in the updated methods section below.

New text:

“Genomic Sequencing and annotation of *Bacillus amyloliquefaciens*

Genomic DNA from *Bacillus amyloliquefaciens* BK WT and transposon mutant Tn5-3 was isolated using PureLink Microbiome DNA purification kit (Invitrogen) according to the manufacturer’s instructions.

The *Bacillus amyloliquefaciens* BK WT genome was assembled by mapping its sequencing data onto the annotated genome of *Bacillus amyloliquefaciens* FZB42 using Geneious Prime. Differences in the mutant strain Tn5-3 were identified by mapping its sequencing data onto the assembled *Bacillus amyloliquefaciens* BK WT genome. The mutated genes were then annotated using NCBI BLAST. The oxydifficidin BGC was annotated using the antiSMASH online server.” (Line 253-260)

(7) Please describe in the methods how you screened the library for strains that lacked anti-gonococcal activity.

The method is added to our revised manuscript as section “Screening of Bacillus Strains Lacking Anti-*N. gonorrhoeae* Activity”.

New text:

“Screening of Bacillus Strains Lacking Anti-N. gonorrhoeae Activity

The transposon mutants of *Bacillus amyloliquefaciens* BK were grown overnight in LB medium at 30 °C. Each overnight culture was then diluted 1:5000, and 1 μl of the diluted culture was spotted onto a GCB agar plate swabbed with *N. gonorrhoeae* cells. The plate was then incubated overnight at 37 °C with 5% CO2. The mutant strain (Tn5-3) lacking anti-*N. gonorrhoeae* activity was identified due to its failure to produce a zone of growth inhibition in the resulting *N. gonorrhoeae* lawn.” (Line 341-346)

(8) Was only one strain found that was a 'non-producer' of anti-N. gonorrhoeae activity? Line 68 suggests that this was only one of multiple non-producers. Is that correct? If so, did you work up the others, and did they also have disruptions in the same biosynthetic gene cluster?

Only one strain was identified as a “non-producer” of anti-*N. gonorrhoeae* activity. We have modified the text to clarify this point.

Original text: “The sequencing of one non-producer strain revealed that it surprisingly contained four transposon insertions and one frame shift mutation.”

Changed to: “The sequencing of the non-producer strain revealed that it surprisingly contained four transposon insertions and one frame shift mutation.” (Line 53-54)

(9) All sequences (including Bacillus amyloliquefaciens BK) must be deposited in a public database (e.g., NCBI) and the accession numbers reported in the manuscript.

Genomic sequence data of *Bacillus amyloliquefaciens* BK has been deposited in GenBank, and its accession number (GCA_019093835.1) now appears in figure legend of Figure S1a.

Figure S1a legend:

“Genome-based phylogenetic tree containing *Bacillus amyloliquefaciens* BK and closely related *Bacillus* spp. The tree was built by Genome Clustering of MicroScope using neighbor-joining method. The NCBI accession numbers of *Bacillus* strains used in the tree are GCA_000196735.1, GCA_000204275.1, GCA_000015785.2, GCA_019093835.1, GCA_000009045.1, GCA_000011645.1, GCA_000172815.1, GCA_000008005.1, and GCA_000007845.1 (from top to bottom).”

Minor(10) Statements in the article would benefit from fact-checking. For example:gonorrhea is not the second most prevalent sexually transmitted infection worldwide; it is the second most reported bacterial sexually transmitted infection.Treatment is ceftriaxone 500mg IM x1 in the US, but 1g IM x1 in the UK and Europe. The UK guidelines also permit ciprofloxacin, should sequencing indicate gyrA 91S. I suggest reviewing / specifying which treatment guidelines you're referring to.

We appreciate the reviewer’s corrections. The word “prevalent” is now changed to “reported”.

Original text: “Gonorrhea, which is caused by *Neisseria gonorrhoeae*, is the second most prevalent sexually transmitted infection worldwide.”

Changed to: “Gonorrhea, which is caused by *Neisseria gonorrhoeae*, is the second most reported sexually transmitted infection worldwide.” (Line 2-3)

Original text: “Gonorrhea is the second most prevalent sexually transmitted infection worldwide, its causative agent is the bacterium *Neisseria gonorrhoeae*.”

Changed to: “Gonorrhea is the second most reported sexually transmitted infection worldwide, its causative agent is the bacterium *Neisseria gonorrhoeae*.” (Line 18-19)

“In the USA” is now added to the sentence stating gonorrhea treatment.

Original text: “The high dose (500 mg) of the cephalosporin ceftriaxone is currently the only recommended therapy for treating gonorrhea infections.”

Changed to: “The high dose (500 mg) of the cephalosporin ceftriaxone is currently the only recommended therapy for treating gonorrhea infections in the USA.” (Line 20-22)

(11) Please make sure all results are in the results section. The report of cell morphology, for example, should be in the results, not the discussion.

In our revised manuscript, we have included the cell morphology data in the results section with the text changes below.

Original text: “Interestingly, not only was *dedA* deficient N. gonorrhoeae less susceptible to oxydifficidin, oxydifficidin also kills this mutant more slowly (Figure 2b) than WT *N. gonorrhoeae* MS11.”

Changed to: “Interestingly, not only was *dedA* deficient N. gonorrhoeae less susceptible to oxydifficidin, oxydifficidin also kills this mutant more slowly (Figure 2b) than WT *N. gonorrhoeae* MS11. The *dedA* deletion mutant also showed an altered cell morphology with reduced membrane integrity and lower formation of micro-colonies (Figure S4). (Line 100-104)

Original text: “The *dedA* deletion mutant also showed an altered cell morphology with reduced membrane integrity and lower formation of micro-colonies (Figure S4), indicating that it should show reduced pathogenesis and fitness, and, as a result, not accumulate in a clinical setting, which adds to the therapeutic appeal of oxydifficidin.”

Changed to: “The *dedA* deletion mutant exhibited altered cell morphology, characterized by diminished membrane integrity and reduced micro-colony formation, indicating that it should show reduced pathogenesis and fitness, and, as a result, not accumulate in a clinical setting, which adds to the therapeutic appeal of oxydifficidin” (Line 206-210)

(12) Tables 1 and 2 should be combined and should address the most relevant antibiotics

The MIC data of additional relevant antibiotics are now included in Table 1. However, we still believe that keeping Tables 1 and 2 separate enhances the clarity of the manuscript. Table 2 specifically focuses on diverse ribosomal targeting antibiotics, which highlights the unique binding site of oxydifficidin.

(13) Supplemental Figure 1a. The tree could be better resolved, and there are four entries with the identical listing of "Bacillus amyloliquefaciens subsp. plantarum" on different branches. In the methods or the legend, please indicate the accession numbers for these genomes. Also please specify how this tree was made-is it a maximum likelihood tree? Something else?

The tree is now better resolved and includes new entries. The requested information regarding accession numbers and tree construction method has been included in the figure legend.

New supplemental Figure 1a legend:

“a. Genome-based phylogenetic tree containing *Bacillus amyloliquefaciens* BK and closely related *Bacillus* spp. The tree was built by Genome Clustering of MicroScope using neighbor-joining method. The NCBI accession numbers of *Bacillus* strains used in the tree are GCA_000196735.1, GCA_000204275.1, GCA_000015785.2, GCA_019093835.1, GCA_000009045.1, GCA_000011645.1, GCA_000172815.1, GCA_000008005.1, and GCA_000007845.1 (from top to bottom).”

**Reviewer #2 (Recommendations For The Authors):**
The conclusions drawn in the manuscript are well-supported by the experimental data presented.I have the below minor comments:(1) "serendipitously identified" - I feel this wording should be avoided throughout the manuscript. The point of a research paper is to communicate methodology and experimental detail, and this language portrays the opposite.

While we agree that methodology and experimental procedures are paramount in scientific reporting, we believe it is equally important to convey, particularly to younger generations, that a part of the scientific process is often unplanned and can benefit from chance observations. Therefore, we would like to keep this wording.

(2) The introduction should include the biological roles/function of DedA proteins in bacteria.

DedA proteins perform a wide array of biological roles and functions in bacteria. In the results section (Line 107-116), we have described the most well-established of these functions, particularly the flippase activity, which appears to be directly related to oxydifficidin sensitivity. We believe that introducing this information in the results section enhances the manuscript’s clarity and flow.

(3) "When we screened this contaminant for antibacterial activity against lawns of other Gram-negative bacteria it did not produce a zone of growth of inhibition against any of the bacteria we tested (e.g., *Escherichia coli*, *Vibrio cholerae*, Caulobacter crescentus)." Can these data Figures be included in the Supplements?

This result was recorded in the lead author’s notebook, but no image was saved.

(4) Line 52: Was any base analyses performed on the Tn-mutants i.e., how many insertion-sites? Depth of mutants? Was a library constructed in this study or previously? Why were only BGC assessed?

Please see our response to Reviewer #1’s comment (5). We focused on BGCs because we believed the anti-*N. gonorrhoeae* activity most likely resulted from a molecule encoded by a natural product BGC.

(5) Line 98: Do the other 2 predicted DedA-like proteins also have a role in uptake of oxydifficidin? Is there some redundancy in uptake?

We generated knockout mutants for two other predicted DedA-like proteins in *N. gonorrhoeae* MS11, and the MIC of oxydifficidin for these mutants remained the same as for the *N. gonorrhoeae* MS11 wild type strain. Therefore, we believe that the DedA protein discussed in this manuscript is the primary transporter of oxydifficidin. However, we cannot completely rule out the possibility of redundancy in oxydifficidin uptake by other DedA-like proteins.

New text: “We also generated deletion mutants for two other predicted *dedA*-like genes, and the MIC of oxydifficidin for these mutants remained the same as for the *N. gonorrhoeae* MS11 wild type strain.” (Line 98-100)

**Reviewer #3 (Recommendations For The Authors):**
This is a well presented manuscript and I could not immediately see any issues with it.

We appreciate the reviewer’s positive feedback.